# A Very Big Video Reasoning Suite

**Maijunxian Wang** [* 1]  **Ruisi Wang** [* 2]  **Juyi Lin** [* 3]  **Ran Ji** [* 4]  **Thaddäus Wiedemer** [5]  **Qingying Gao** [6]  **Dezhi Luo** [7]
**Yaoyao Qian** [3]  **Lianyu Huang** [8]  **Zelong Hong** [9]  **Jiahui Ge** [8]  **Qianli Ma** [10]  **Hang He** [11]  **Yifan Zhou** [10]
**Lingzi Guo** [12]  **Lantao Mei** [12]  **Jiachen Li** [13]  **Hanwen Xing** [8]  **Tianqi Zhao** [14]  **Yu Fengyuan** [2]  **Weihang Xiao** [15]
**Yizheng Jiao** [16]  **Jianheng Hou** [8]  **Danyang Zhang** [17]  **Pengcheng Xu** [18]  **Boyang Zhong** [19]  **Zehong Zhao** [4]
**Gaoyun Fang** [20]  **John Kitaoka** [21]  **Xu Yile** [22]  **Hua Xu** [23]  **Kenton Blacutt** [24]  **Tin Nguyen** [25]  **Siyuan Song** [13]
**Haoran Sun** [6]  **Shaoyue Wen** [20]  **Linyang He** [26]  **Runming Wang** [6]  **Yanzhi Wang** [3]  **Mengyue Yang** [27]  **Ziqiao Ma** [7]
**Raphaël Millière** [28]  **Freda Shi** [29]  **Nuno Vasconcelos** [4]  **Daniel Khashabi** [6]  **Alan Yuille** [6]  **Yilun Du** [30]  **Ziming Liu** [12]
**Dahua Lin** [31]  **Ziwei Liu** [2]  **Vikash Kumar** [32]  **Yijiang Li** [4]  **Lei Yang** [31]
**Zhongang Cai** [✉ 2]  **Hokin Deng** [✉ 32]

## Abstract

Rapid progress in video models has largely focused on visual quality, leaving their reasoning capabilities underexplored. Video reasoning grounds intelligence in spatiotemporally consistent visual environments that go beyond what text can naturally capture, enabling intuitive reasoning over spatiotemporal structure, such as continuity, interaction, and causality. However, systematically studying video reasoning and its scaling behavior is hindered by the lack of large-scale video reasoning training data. To address this gap, we introduce the **Very Big Video Reasoning (VBVR) Dataset**, an unprecedentedly large-scale resource spanning *150* curated reasoning tasks following a principled taxonomy, and over *one million* video clips, making it approximately *three orders of magnitude* larger than existing datasets. We further present **VBVR-Bench**, a verifiable evaluation framework that moves beyond model-based judging by incorporating rule-based, human-aligned scorers, enabling reproducible and interpretable diagnosis of video reasoning capabilities. Lever-

aging the VBVR suite, we conduct one of the first video reasoning **scaling studies** and observe early signs of emergent generalization to unseen reasoning tasks. Together, VBVR lays a foundation for the next stage of research in generalizable video reasoning. The data, benchmark tool kit, and models are released publicly at **video-reason.com**.

## 1. Introduction

Ground-breaking progress has been achieved in large language models, whose reasoning abilities now generalize across challenging tasks such as coding, mathematics, and scientific discovery (Mitchell, 2025). However, such capabilities remain largely confined to text-based scenarios. Meanwhile, recent advances in video generation models have predominantly emphasized visual realism, with comparatively limited focus on reasoning capabilities. Yet video models hold the potential to support a new paradigm of reasoning (Wiedemer et al., 2025), grounded in spatiotemporally consistent visual environments where spatial structure, physical dynamics, and long-range causality are naturally encoded. This makes video frames an ideal substrate for studying reasoning grounded in the physical world. Despite this promise and growing interest in video reasoning[1], the community still lacks several critical components required for systematic progress: (1) a large-scale and diverse **dataset** to enable meaningful investigation of scaling and generalization, (2) an **evaluation toolkit** built on verifiable and reproducible principles, and (3) an initial **scaling study** that

---
[*]Equal contribution  [1]UC Berkeley  [2]NTU  [3]Northeastern Univ.  [4]UC San Diego  [5]Univ. of Tübingen  [6]Johns Hopkins Univ.  [7]Univ. of Michigan  [8]USC  [9]Washington Univ. in St. Louis  [10]Shanghai Jiao Tong Univ.  [11]East China Normal Univ.  [12]Stanford Univ.  [13]UT Austin  [14]UCLA  [15]Cornell Univ.  [16]UNC Chapel Hill  [17]San Jose State Univ.  [18]UC Irvine  [19]TU Munich  [20]Imperial College London  [21]UW–Madison  [22]Univ. of Edinburgh  [23]HKUST  [24]NYU  [25]Auburn Univ.  [26]Columbia Univ.  [27]Univ. of Bristol  [28]Univ. of Oxford  [29]Univ. of Waterloo  [30]Harvard Univ.  [31]CUHK  [32]CMU. Correspondence to: Hokin Deng <hokind@andrew.cmu.edu>, Zhongang Cai <caiz0023@e.ntu.edu.sg>.

*Proceedings of the 43rd International Conference on Machine Learning*, Seoul, South Korea. PMLR 306, 2026. Copyright 2026 by the author(s).

---
[1]We use "video reasoning" in the broader sense of *reasoning through video*, where video is the substrate (rather than only the input) of intelligence; this includes generating spatiotemporally consistent outputs that satisfy physical and logical constraints, aligning with the emerging *generation-as-reasoning* paradigm (Chen et al., 2025; Yang et al., 2025a; Wiedemer et al., 2025).

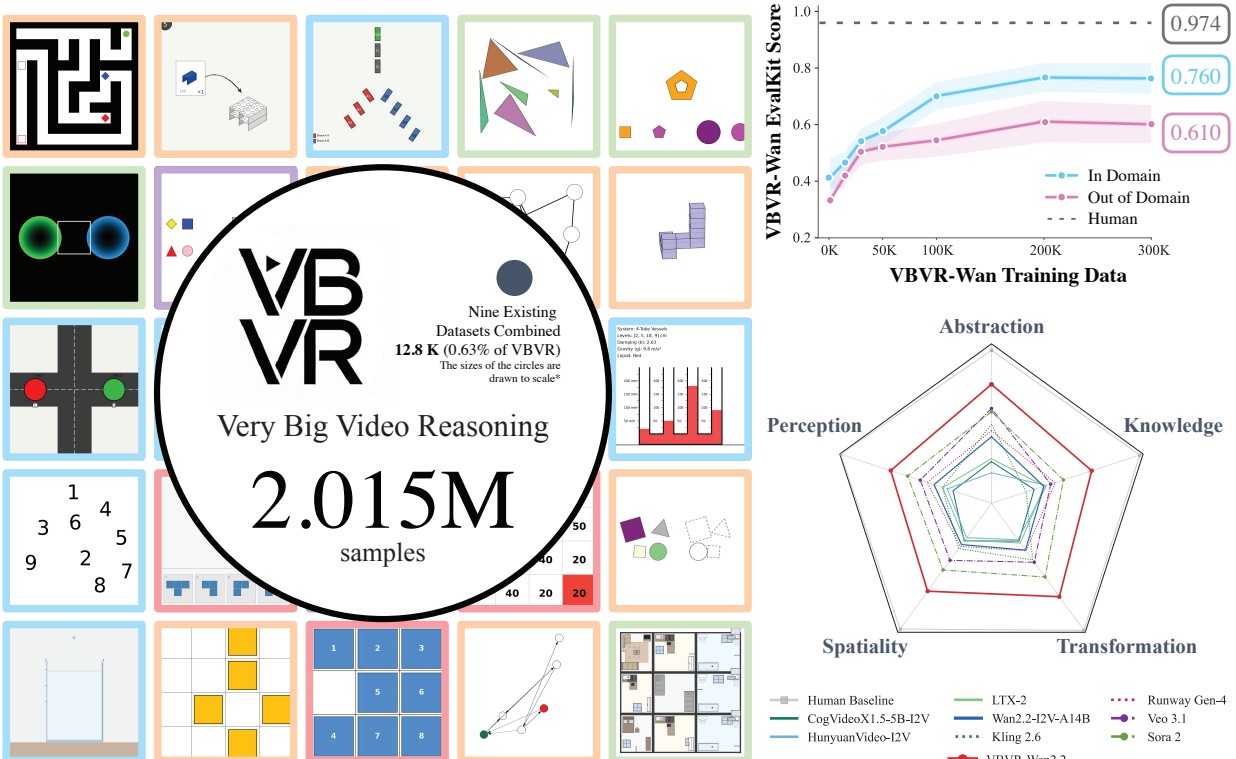

*Figure 1.* **Overview of VBVR.** Left: the grid shows representative tasks spanning our cognitive taxonomy, which are color-coded according to their underlying capability: Spatiality, Transformation, Knowledge, Abstraction, and Perception. At the center of the grids, we visualize the scale comparison between VBVR (2.015M samples) and nine other datasets combined (12.8K samples): the sizes of the circles are drawn to scale. Top-right: scaling behavior on in-domain and out-of-domain evaluations. Bottom-right: benchmark performance across five cognitive capabilities.

examines emergent capabilities in video reasoning models. In this work, we address all three challenges by introducing the **Very Big Video Reasoning (VBVR)** suite (Figure 1).

First, we introduce **VBVR-Dataset**, a large-scale and diverse training source designed for systematic study of video reasoning. We adopt a principled approach, grounding our task taxonomy in well-established constructs from cognitive and developmental psychology (Spelke & Kinzler, 2007; Carey, 2009; Anderson, 2007). Specifically, we organize core visual reasoning capabilities into five dimensions: *perception*, *transformation*, *spatiality*, *abstraction*, and *knowledge*. The dataset is a collaborative effort involving over 50 researchers and engineers from diverse disciplines worldwide, ensuring broad coverage and strong domain expertise across 150 tasks to date. Contributors are given full freedom to design the task semantics and reasoning procedures, allowing for maximal diversity, while a unified, overarching task template is applied as a standardized wrapper for input and output specification. This separation ensures consistency for automated scaling without constraining task creativity. All tasks undergo expert human inspection to ensure quality and correctness before being processed by our automated, cloud-based pipeline, which generates randomized training examples at scale in a distributed manner. In total, VBVR-Dataset contains 2,015,000 images and 1,007,500

*Table 1.* **Comparison of VBVR-Dataset with existing video reasoning benchmarks.** VBVR-Dataset surpasses prior benchmarks by multiple orders of magnitude across every dimension and is the first to provide large-scale training data for video reasoning.

| Dataset | #Task | #Images | #Videos | #Train | #Test |
|---|---|---|---|---|---|
| Video-Zero-Shot (Wiedemer et al., 2025) | 69 | 1,578 | 0 | 0 | 2,128 |
| V-ReasonBench (Luo et al., 2025) | 13 | 652 | 0 | 0 | 326 |
| MMGR (Cai et al., 2025) | 10 | 1,323 | 530 | 0 | 1,853 |
| VideoThinkBench (Tong et al., 2026) | 24 | 8,298 | 0 | 0 | 4,149 |
| TiViBench (Chen et al., 2025) | 24 | 595 | 0 | 0 | 595 |
| VR-Bench (Yang et al., 2025a) | 5 | 0 | 7,920 | 6,336 | 1,538 |
| MME-CoF (Guo et al., 2025) | 12 | 120 | 0 | 0 | 120 |
| Gen-ViRe (Liu et al., 2025) | 24 | 117 | 0 | 0 | 72 |
| Ruler-Bench (He et al., 2025) | 40 | 101 | 0 | 0 | 622 |
| **VBVR-Dataset** | **150** | **2,015,000** | **1,007,500** | **1,000,000** | **7,500** |

video clips, making it roughly 1,000× larger than existing alternatives. Importantly, the pipeline is immediately compatible with newly added tasks and supports scalable generation of additional examples per task, enabling continuous expansion in both dataset breadth and scale.

Second, **VBVR-Bench** provides a systematic, reproducible, and explainable evaluation framework for video reasoning models. While VLM-as-a-judge paradigms have been widely adopted for evaluating video generation models (Peng et al., 2025), we enforce the use of verifiable, rule-based scorers to ensure transparent and reproducible evaluation. To validate that these task-specific scorers faithfully reflect model capabilities, we conduct human preference alignment experiments, observing strong agreement between automated scores and human judgments, with

Spearman's correlation coefficient $\rho > 0.9$. Leveraging VBVR-Bench, we benchmark leading proprietary models: Veo 3.1 (Google DeepMind, 2026), Sora 2 (OpenAI, 2025), Kling 2.6 (Kuaishou Technology, 2025), and Runway Gen-4 (Runway Research, 2025), alongside representative open-source models, including Wan2.2 (WanTeam, 2025), CogVideoX-1.5 (Yang et al., 2025b), HunyuanVideo (Kong et al., 2025), and LTX-2 (HaCohen et al., 2026). We reveal a substantial gap in video reasoning capabilities across systems; the strongest model falls short of human performance by a large margin. We further analyze how cognitive capabilities co-develop across models, uncovering non-trivial dependencies and trade-offs among reasoning skills.

Third, with a large-scale dataset and a reliable evaluation benchmark in place, we conduct an in-depth investigation of scaling effects in video generation models. Using Wan2.2 as the base model, we observe concurrent improvements on both in-domain (ID) and out-of-domain (OOD) tasks as training scale increases, indicating the gradual emergence of generalization capabilities. Beyond these gains, our analysis yields several key insights. First, performance on both ID and OOD tasks eventually plateaus as data scale increases, leaving a persistent gap between model and human performance that data scaling alone cannot close. This suggests fundamental limitations in current video generation architectures when applied to video reasoning. Second, despite substantial OOD gains with scale, a consistent gap remains between ID and OOD settings; narrowing this gap appears essential for robust, in-the-wild video reasoning and generation. Finally, qualitative analyses reveal emergence in instruction following, controlled editing, and semantic understanding with increased data scale, while also exposing important limitations that motivate future research.

In summary, we present the VBVR suite, centered on an unprecedentedly large-scale and continually growing dataset for video reasoning, **VBVR-Dataset**, together with a verifiable, human-aligned evaluation toolkit, **VBVR-Bench**. Leveraging this suite, we conduct one of the first systematic scaling studies of video reasoning models and uncover early, encouraging evidence of emergent generalization. We believe VBVR provides a foundational infrastructure for future research toward generalizable video reasoning.

**Conflict of Interest Disclosure.** The authors declare no financial conflicts of interest related to the models evaluated in this work. None of the authors is employed by, holds equity in, or receives consulting fees from organizations developing the video generation systems benchmarked.

## 2. Related Works

**Video Generation Models.** Since the inauguration of diffusion models and transformer-based scaling (Ho et al., 2020; Peebles & Xie, 2023), video generation models are proliferating, with closed models such as Sora, MovieGen, and Veo, and open-source ones like CogVideoX, HunyuanVideo, and Wan (OpenAI, 2025; Polyak et al., 2024; Google DeepMind, 2026; Yang et al., 2025b; Kong et al., 2025; WanTeam, 2025). But many models are optimized for creative production rather than explicit relational, causal, or counterfactual reasoning (Peebles & Xie, 2023; Yang et al., 2025b; Zheng et al., 2024), leaving open the question of whether they learn structured world models needed to support spatiotemporal reasoning (LeCun, 2022).

**Generation-as-Reasoning.** Recent research increasingly investigates video generation not only as a content-creation tool, but as a reasoning substrate (Tong et al., 2026; Guo et al., 2025; Liu et al., 2025; Wiedemer et al., 2025). A recent study has tested Veo-3 and shown early evidence that the model exhibits nontrivial zero-shot perceptual and manipulation behaviors and can solve simple tasks without task-specific training (Wiedemer et al., 2025). Later works now span generation-as-reasoning paradigms (Tong et al., 2026), multi-step Chain-of-Frame diagnosis (Guo et al., 2025; Liu et al., 2025), TI2V answer suites (Luo et al., 2025; Chen et al., 2025), among others (He et al., 2025; Yang et al., 2025a; Cai et al., 2025). Despite sharper measurement, the ecosystem remains evaluation-heavy: large-scale training splits and ablation protocols are missing, making it difficult to run reproducible scaling studies that optimize for reasoning correctness.

**Procedural Generation vs. MLLM-driven Curation.** One way to supply such training-scale supervision is to scale multimodal annotation through MLLMs on existing image–text pairs (e.g., Bee (Zhang et al., 2025)). VBVR instead derives deterministic ground truth from manually implemented simulators, providing physically grounded supervision over video that can be checked by rule-based, task-specific evaluators (Section 4.1) rather than VLM judges.

## 3. Dataset

In this section, we describe the cognitive taxonomy underlying our systematic task design (Section 3.1), present the key statistics of VBVR-Dataset (Section 3.2), and detail the data generation pipeline (Section 3.3).

### 3.1. Cognitive Taxonomy

Systematically evaluating video reasoning requires a principled taxonomy of the cognitive capabilities under test. We organize VBVR around five capability dimensions, drawing on well-established constructs in cognitive and developmental psychology (Newell & Simon, 1972; Anderson, 2007; Spelke & Kinzler, 2007; Carey, 2009). **Perception** refers to the extraction of structured representations from sen-

sory input—edge detection, shape and color discrimination, figure–ground segmentation—grounded in the hierarchical architecture of the visual system (Hubel & Wiesel, 1962; DiCarlo et al., 2012; Marr, 1982). **Transformation** is the manipulation and synthesis of mental representations, operationalized through tasks such as mental rotation, reflection, and recombination (Shepard & Metzler, 1971; Zacks, 2008; Kosslyn, 1994). **Spatiality** is the representation of places and their geometric relationships, probed through navigation and spatial reasoning tasks, grounded in research on cognitive maps and spatial cognition (Tolman, 1948; O'Keefe & Dostrovsky, 1971; Hafting et al., 2005). **Abstraction** is the extraction of generalizable patterns from particular instances—tested through tasks like Raven's Progressive Matrices and sequence completion—drawing on theories of analogical reasoning and conceptual development (Carey, 2009; Gentner, 1983; Badre & Nee, 2018). **Knowledge** encompasses domain-specific representational systems that structure reasoning about the physical and symbolic world, including intuitive physics (object permanence, solidity, causality) and learned symbolic knowledge (numerical comparison, symbol manipulation), grounded in the core knowledge framework (Spelke & Kinzler, 2007; Spelke, 2000; Baillargeon et al., 1985; Li et al., 2025). These dimensions are not strictly independent—our capability correlation analysis (Section C.6) reveals systematic dependencies between them—but each targets a sufficiently distinct family of cognitive operations to support meaningful diagnostic evaluation. By using procedurally generated, visually abstracted stimuli rather than naturalistic video, VBVR isolates spatiotemporal reasoning from confounds such as texture bias, camera variation, and training-set memorization, following the same logic that motivates benchmarks like ARC (Chollet, 2019) and CLEVR (Johnson et al., 2017) in other domains. To operationalize these dimensions in a video-based reasoning setting, VBVR implements each category as a family of parameterized task generators. Representative task instances for each dimension are illustrated in Figure 2. Detailed cognitive science grounding and neuroscientific evidence for each dimension appear in Section A.

### 3.2. Data Statistics

Table 1 compares VBVR-Dataset with existing video reasoning benchmarks. VBVR-Dataset surpasses prior work by *multiple orders of magnitude* across all key dimensions. Notably, most existing video reasoning benchmarks provide few or no video samples (often lacking training data altogether), which has been a major bottleneck for studying scaling. In total, VBVR-Dataset comprises 150 publicly released reasoning tasks.

### 3.3. Data Curation

The curation pipeline has three stages: **(1) Task design and approval**, where each proposal must satisfy six quality criteria (information sufficiency, deterministic solvability, video dependency, visual clarity, parametric diversity, and technical feasibility); 150 designs have been approved from over 500 proposals. **(2) Task-specific generator implementation**, where each approved design becomes a parameterized generator inheriting from a standardized template, deterministically emitting a 4-tuple (`first_frame.png`, `prompt.txt`, `final_frame.png`, `ground_truth.mp4`) so that components 1–2 serve as model input and 3–4 provide verifiable supervision over the full reasoning trajectory. **(3) Large-scale generation and control**, where validated generators run in a distributed framework to produce 1M training samples (100 tasks × 10K) and 7,500 test samples (150 tasks × 50), with disjoint seed ranges across splits, automated visual-compliance / boundary / edge-case validation, automatic retries, and per-task failure logging (Section B.3.1). The infrastructure supports community expansion; implementation details are provided in Section B.

## 4. Benchmark

In this section, we introduce the evaluation toolkit (Section 4.1) and assess its validity through alignment with human preferences (Section 4.2). We subsequently report the performance of leading video generation models (Section 4.3); a structural analysis of how reasoning capabilities co-develop is provided in Section C.6.

### 4.1. Evaluation Kit

To systematically assess model reasoning capabilities, VBVR-Bench employs a dual-split evaluation strategy across 100 diverse tasks. The first split contains 50 tasks that overlap with the training categories but differ in unseen parameter configurations and sample instances, providing a test of *in-domain generalization*. The second split includes the remaining 50 tasks, which are entirely novel and are designed to measure *out-of-domain generalization*. It tests whether models can solve reasoning challenges without prior exposure to similar structures, and thus whether they acquire transferable reasoning primitives rather than relying on task-specific memorization. Each task consists of five test samples, enabling statistically robust evaluation across diverse reasoning scenarios.

A key feature of VBVR-Bench is its fully **rule-based evaluation framework**, which is feasible because most test tasks have a unique, verifiable correct answer, allowing interpretable evaluation based on spatial position, color, object identity, path, or logical outcome. Moreover, geometric,

*Figure 2.* **VBVR task examples.** Sample task instances generated from the VBVR parameterized task suite, organized by five cognitive dimensions. Each sequence illustrates the structured reasoning process required to reach a valid solution. Tasks are implemented as deterministic generators supporting scalable instance variation while preserving visual clarity and video dependency. Each task is labeled with its cognitive capability dimension (Section 3.1) and instantiated as an executable, verifiable video-based reasoning task.

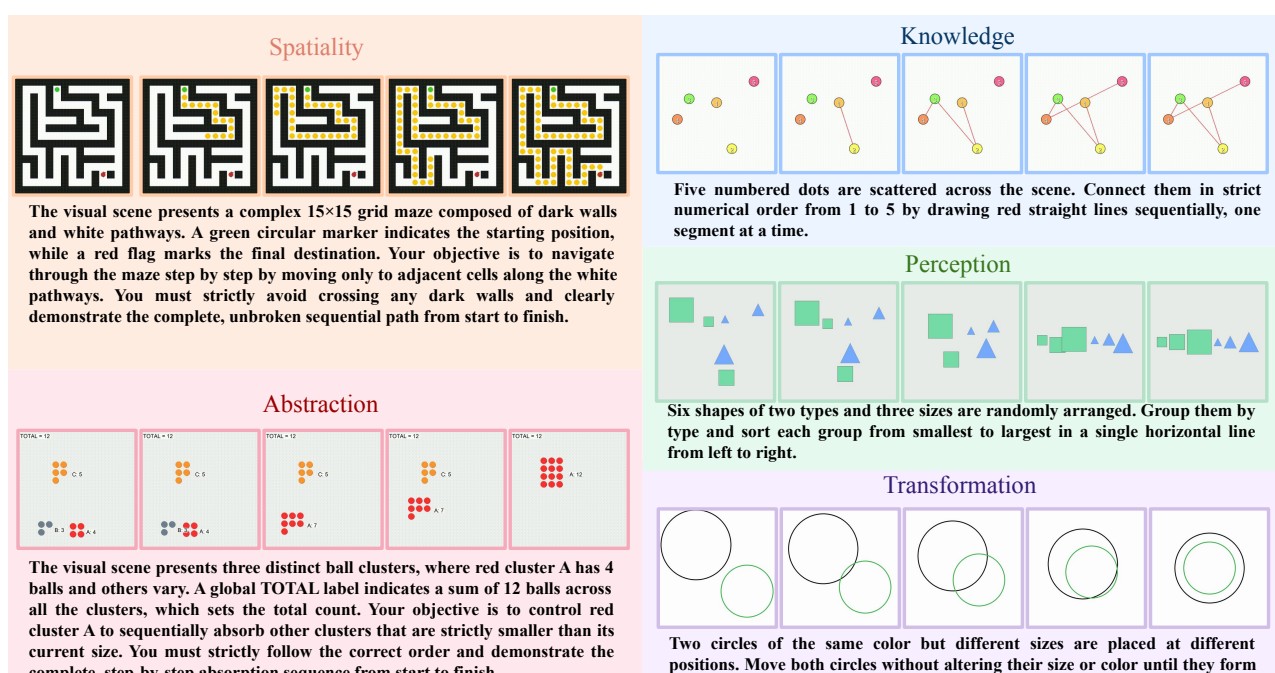

physical, or deductive constraints are also considered in the scoring rubrics. Each of the 100 test tasks is paired with a dedicated evaluation rule, with scores on multiple aspects to compute a weighted, comprehensive score. Sub-criteria include spatial accuracy, trajectory correctness, temporal consistency, and logical validity.

For example, in Task G-45: Key Door Matching (more examples are included in Section D), a colored-dot agent must locate a key and navigate to a same-colored door within a grid maze. Performance is scored across four weighted dimensions: **agent movement** (30%), **key collection** (35%), **door reached** (25%), and **sequence** (10%). Agent movement requires the agent to physically traverse the maze away from its starting position; key collection requires the agent to reach a key's location *and* for that key to disappear thereafter; door reached requires the agent to subsequently arrive at a door whose color matches the collected key; and sequence checks that the key is collected before the door is reached. Pseudocode for this rule is provided in Section C.7.

**Disentangling Reasoning from Generative Quality.** A natural question is whether observed failures stem from limited generative capacity (texture, lighting, motion fidelity) or from genuine reasoning deficits. Complete separation remains open, but our design reduces this confound: synthetic, low-texture scenes minimize perceptual nuisance, and rule-based scorers verify outcomes against deterministic ground truth rather than hallucinating VLM judges. Consistent with

this, recent analysis of VBVR-Wan2.2 (Wang et al., 2026) finds that video diffusion models do not merely memorize: early denoising steps maintain multiple candidate solutions progressively pruned into a consistent outcome—implicit reasoning that becomes observable precisely because evaluation confounds are controlled.

Overall, VBVR-Bench provides three properties: **reproducibility and determinism**—the evaluation is fully deterministic, avoiding the stochastic variability or hallucinations associated with VLM-based judgments; **granular verifiability**: each task is decomposed into interpretable vectors, allowing precise measurement of spatial, temporal, and logical correctness at the pixel or object level; and **transparent diagnosis**: by explicitly encoding reasoning constraints, the benchmark ranks models *and* reveals systematic trade-offs, capability gaps, and cross-domain performance trends.

### 4.2. Human Preference Alignment Analysis

To assess alignment between VBVR-Bench and human perception, we conduct a large-scale human preference study across 9 models and compare model win ratios derived from human judgments with those computed from VBVR-Bench's automatic metrics. Specifically, human win ratios are obtained from pairwise preference annotations, where a model is considered to win if it is preferred over another model for the same prompt. In contrast, VBVR-Bench win ratios are computed by ranking models by their per-sample

automatic scores and counting how often each model outperforms others. As shown in Figure 3, the two sets of win ratios exhibit strong positive correlations across models on both in-domain and out-of-domain splits (Spearman's $\rho > 0.9$), indicating that VBVR-Bench provides reliable and human-aligned performance estimates. Details of the study are provided in Section C.2.

### 4.3. Leading Model Performances

Table 2 shows performance across model families. Most open-source baselines cluster between 0.27 and 0.31 overall, indicating limited capability in complex video reasoning, while Wan2.2-I2V-A14B is the strongest open-source baseline at 0.371. Proprietary models perform better overall, led by Sora 2 (0.546) and Veo 3.1 (0.480), particularly in Abstraction and Transformation categories.

Fine-tuning Wan2.2-I2V-A14B on VBVR-Dataset yields VBVR-Wan2.2, which achieves a new state of the art with an overall score of 0.685, representing an 84.6% relative improvement over its base model. VBVR-Wan2.2 attains the best performance across all evaluated categories, with especially strong results in Spatiality and Perception, suggesting that large-scale reasoning-oriented data substantially enhances integrated world-model reasoning capabilities.

To verify that the gains from VBVR fine-tuning are not specific to the Wan2.2 backbone, we additionally apply the same procedure to LTX-2.3 (HaCohen et al., 2026), a structurally distinct dual-stream transformer that jointly generates audio and video via cross-attention between the streams. The resulting VBVR-LTX2.3 achieves an overall score of 0.516, surpassing several proprietary baselines including Runway Gen-4 Turbo, Kling 2.6, and Veo 3.1, and exhibits the same saturating data-scaling pattern as VBVR-Wan2.2 (full scaling trajectory in Table 9). This confirms that VBVR data benefits multiple backbone architectures, while the residual gap between VBVR-LTX2.3 and VBVR-Wan2.2 underscores that backbone choice also affects absolute performance.

Notably, despite these gains, a considerable gap to human performance remains. This highlights the persistent challenges of long-horizon temporal reasoning and robust symbolic manipulation in video generation. We further analyze how performance evolves with increasing training data under a fixed architecture, and how in-domain and out-of-domain generalization behaviors differ, in Section 5.

We also analyze the stability and consistency of model behavior using domain-wise score distributions that reveal performance variability and rating noise across domains (see Section C.3.2). Beyond mean performance, a residualized analysis of how the five cognitive capabilities co-develop—revealing structured trade-offs such as Knowl-

edge coupling positively with Spatiality yet negatively with Perception—is provided in Section C.6.

## 5. VBVR-Wan2.2 Analysis

Among the evaluated systems, VBVR-Wan2.2 achieves the strongest overall performance (Table 2). We therefore conduct an in-depth analysis on this model to gain insights into scaling video reasoning, covering experimental settings (Section 5.1), scaling behavior (Section 5.2), comprehensive qualitative evaluations (Section 5.3), and performance on general video benchmarks (Section 5.4).

### 5.1. Experiment settings

We conduct all experiments on Wan2.2-I2V-A14B without architectural modifications, as the goal of VBVR-Wan2.2 is to **investigate data scaling behavior** and provide a **strong baseline model** for the video reasoning research community. Leveraging the VBVR-Dataset, which to our knowledge constitutes one of the largest video reasoning datasets to date, enables a systematic investigation of scaling behaviors in video-based reasoning under a fixed model architecture. For training, we adopt a learning rate of 1e-4 and train for one epoch in each experiment. We employ LoRA adaptation on the DiT backbone, applying it to the modules q, k, v, o, ffn.0, ffn.2 with a LoRA rank of 32.

### 5.2. Scaling Curve

To investigate data scaling behavior, we progressively increase the training data size from 0K samples (the original Wan2.2 base model) to 500K samples (VBVR-Wan2.2), and report the performance changes in Table 3.

First, training VBVR-Wan2.2 shows clear but **saturating gains from data scaling**. In-domain (ID) performance improves substantially with increased training data, rising from 0.412 at initialization to about 0.771 at 400K samples, after which gains plateau and slightly fluctuate. The failure to approach perfect accuracy, even within familiar distributions, suggests that current video generation architectures exhibit fundamental representational and optimization bottlenecks. In particular, these tasks require the simultaneous satisfaction of logical constraints and long-term temporal consistency, while the stochastic nature of video generation introduces cumulative rendering noise and temporal drift. Importantly, this saturation regime makes VBVR-Dataset a valuable testbed for researchers to investigate architectural advances, such as explicit state tracking, structured reasoning modules, or self-correction mechanisms, under controlled and scalable evaluation settings.

Second, examining generalization to **out-of-domain (OOD) tasks** highlights critical insights. Both ID and OOD performance improve with more data, ID from 0.412 to 0.760,

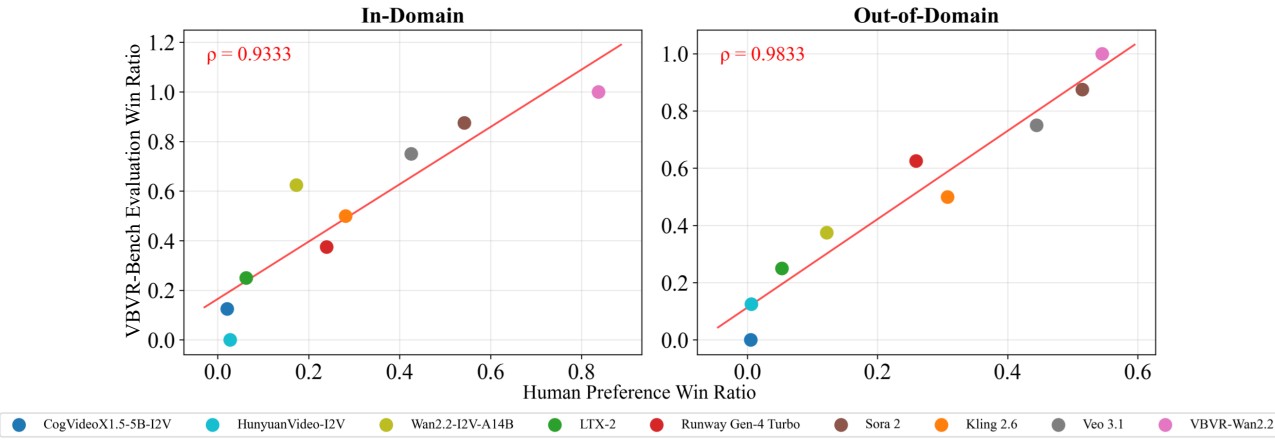

*Figure 3.* **Human alignment analysis for VBVR-Bench.** Our experiments show that VBVR-Bench evaluations in all splits closely match human perceptions. In each plot, a dot represents the human preference win ratio (horizontal axis) and VBVR-Bench evaluation win ratio (vertical axis) for a particular video generation model. We linearly fit a straight line to visualize the correlation, and calculate the Spearman's correlation coefficient ($\rho$) for each dimension.

*Table 2.* **Benchmarking results on VBVR-Bench.** Overall In-Domain (ID) and Out-of-Domain (OOD) scores are reported alongside category-wise performance. Higher is better. **Bold**: best in group; underline: second best.

| Models | Overall | Avg. | In-Domain by Category | | | | | Avg. | Out-of-Domain by Category | | | | |
| --- | --- | --- | --- | --- | --- | --- | --- | --- | --- | --- | --- | --- | --- |
| | | | Abst. | Know. | Perc. | Spat. | Trans. | | Abst. | Know. | Perc. | Spat. | Trans. |
| **Human** | 0.974 | 0.960 | 0.919 | 0.956 | 1.00 | 0.95 | 1.00 | 0.988 | 1.00 | 1.00 | 0.990 | 1.00 | 0.970 |
| **Open-source Models** | | | | | | | | | | | | | |
| CogVideoX1.5-5B-I2V (Yang et al., 2025b) | 0.273 | 0.283 | 0.241 | 0.328 | 0.257 | 0.328 | 0.305 | 0.262 | 0.281 | 0.235 | 0.250 | **0.254** | 0.282 |
| HunyuanVideo-I2V (Kong et al., 2025) | 0.273 | 0.280 | 0.207 | 0.357 | 0.293 | 0.280 | 0.316 | 0.265 | 0.175 | **0.369** | 0.290 | 0.253 | 0.250 |
| Wan2.2-I2V-A14B (WanTeam, 2025) | **0.371** | **0.412** | **0.430** | **0.382** | **0.415** | **0.404** | **0.419** | **0.329** | **0.405** | 0.308 | **0.343** | 0.236 | 0.307 |
| LTX-2 (HaCohen et al., 2026) | 0.313 | 0.329 | 0.316 | 0.362 | 0.326 | 0.340 | 0.306 | 0.297 | 0.244 | 0.337 | 0.317 | 0.231 | **0.311** |
| **Proprietary Models** | | | | | | | | | | | | | |
| Runway Gen-4 Turbo (Runway Research, 2025) | 0.403 | 0.392 | 0.396 | 0.409 | 0.429 | 0.341 | 0.363 | 0.414 | 0.515 | 0.429 | 0.419 | 0.327 | 0.373 |
| Sora 2 (OpenAI, 2025) | **0.546** | **0.569** | 0.602 | 0.477 | **0.581** | **0.572** | **0.597** | **0.523** | 0.546 | **0.472** | **0.525** | **0.462** | **0.546** |
| Kling 2.6 (Kuaishou Technology, 2025) | 0.369 | 0.408 | 0.465 | 0.323 | 0.375 | 0.347 | 0.519 | 0.330 | 0.528 | 0.135 | 0.272 | 0.356 | 0.359 |
| Veo 3.1 (Google DeepMind, 2026) | 0.480 | 0.531 | **0.611** | **0.503** | 0.520 | 0.444 | 0.510 | 0.429 | **0.577** | 0.277 | 0.420 | 0.441 | 0.404 |
| **Data Scaling Strong Baselines** | | | | | | | | | | | | | |
| VBVR-Wan2.2 | **0.685** | **0.760** | **0.724** | **0.750** | **0.782** | **0.745** | **0.833** | **0.610** | **0.768** | **0.572** | **0.547** | **0.618** | **0.615** |
| VBVR-LTX2.3 | 0.516 | 0.580 | 0.608 | 0.631 | 0.529 | 0.454 | 0.680 | 0.453 | 0.608 | 0.577 | 0.409 | 0.414 | 0.388 |

and OOD from 0.329 to 0.610. This indicates that **scaling data enhances transferable reasoning capabilities** beyond memorized patterns. Our qualitative analysis in Section 5.3 further illustrates how the model benefits from increased training data and generalizes to out-of-domain tasks, providing interpretable insights into improvements in temporal consistency, logical reasoning, and task transferability. However, a persistent 15% generalization gap remains, suggesting that increasing data within fixed task distributions is insufficient for robust systematic generalization, consistent with observations that scaling alone does not close the world-modeling gap (Kang et al., 2025).

With our data factory, we plan to continuously introduce new task families and richer compositional regimes in future releases, enabling broader coverage of reasoning patterns and better closing the ID–OOD gap.

### 5.3. Qualitative Analysis

We qualitatively compare Wan2.2-I2V-A14B (base model), VBVR-Wan2.2, and the strongest proprietary baseline in our study, Sora 2. A recurring pattern is that, after VBVR training, VBVR-Wan2.2 can match or even surpass Sora 2 on a broad set of tasks that require *verifiable* manipulations under stable scenes. This motivates our central takeaway: **controllability before reasoning**. If a model freely rewrites the scene (background/layout/object identity) during generation, intermediate states become unreliable and any "reasoning action" (delete/move/mark) is no longer verifiable. In practice, the base model (Wan2.2-I2V-A14B) often fails in precisely this way: it may not preserve target identity or stable layouts, thereby breaking the prerequisite for manipulation-based reasoning.

To make qualitative comparisons direct and reproducible, we select examples as **same-task, same-sample comparisons**

*Table 3.* **Performance by Data Scale (0K–500K) on VBVR-Wan2.2. Bold**: best per column; underline: second best.

| Models | Overall | In-Domain by Category | | | | | | Out-of-Domain by Category | | | | | |
| | | Avg. | Abst. | Know. | Perc. | Spat. | Trans. | Avg. | Abst. | Know. | Perc. | Spat. | Trans. |
|---|---|---|---|---|---|---|---|---|---|---|---|---|---|
| **0K** | 0.371 | 0.412 | 0.430 | 0.382 | 0.415 | 0.404 | 0.419 | 0.329 | 0.405 | 0.308 | 0.343 | 0.236 | 0.307 |
| **50K** | 0.549 | 0.576 | 0.527 | 0.584 | 0.537 | 0.642 | 0.654 | 0.522 | 0.596 | 0.584 | 0.507 | 0.482 | 0.490 |
| **100K** | 0.623 | 0.701 | 0.622 | 0.680 | 0.777 | 0.719 | 0.759 | 0.545 | 0.622 | 0.524 | 0.533 | 0.557 | 0.517 |
| **200K** | **0.689** | 0.767 | 0.739 | 0.709 | 0.791 | **0.799** | 0.825 | **0.611** | 0.748 | **0.621** | 0.545 | **0.659** | 0.599 |
| **300K** | 0.682 | 0.763 | 0.733 | 0.713 | **0.795** | 0.776 | 0.827 | 0.601 | 0.732 | 0.596 | 0.542 | 0.628 | 0.600 |
| **400K** | 0.682 | **0.771** | **0.744** | 0.744 | 0.793 | 0.753 | **0.848** | 0.593 | 0.742 | 0.592 | 0.532 | 0.605 | 0.588 |
| **500K** | 0.685 | 0.760 | 0.724 | **0.750** | 0.782 | 0.745 | 0.833 | 0.610 | **0.768** | 0.572 | **0.547** | 0.618 | **0.615** |

whenever multiple models are shown on the same case. Importantly, the representative cases in Figure 4 are **out-of-domain (OOD) task families held out from training**, so improvements reflect transfer to novel task structures rather than memorization.

**Controllable execution under constraints (VBVR-Wan2.2 vs Sora 2).** Panel A highlights that VBVR training primarily improves *constraint-following, tool-like execution* under stable scenes. On O-5, Sora 2 introduces extra, unnecessary operations: after deleting the target symbol, it further merges/re-layouts the remaining symbols, violating the intended minimal-edit constraint from the task. In contrast, VBVR-Wan2.2 exhibits an emergent "do exactly what is asked" capability, deleting the marked symbol *without* additional changes. On O-6, Sora 2 may fail to maintain scene control and to distinguish the target region from the object to be manipulated, leading to a degenerate outcome where the box and object rotate together. In contrast, VBVR-Wan2.2 correctly separates the target cue from the manipulated object and performs a pivot-based rotation that aligns with the task requirement, suggesting emergent geometric manipulation skills beyond the training task families. On O-30, VBVR-Wan2.2 successfully performs the required constrained relocation (moving a book into the designated slot). Notably, Sora 2 can fail by producing auxiliary markings/lines without executing the actual object manipulation, illustrating that even strong proprietary models may break down when success requires precise, constraint-following control rather than generic scene editing.

**Emergent strategies and multi-step behavior (VBVR-Wan2.2).** Beyond controlled, tool-like execution, we observe emergent *strategy-level* regularities and multi-step behaviors on OOD task families. On O-49, VBVR-Wan2.2 often produces a rule-consistent completion of the missing half while exhibiting a distinctive, *self-chosen* completion policy: across samples, the completion typically appears as a coherent "fade-in" fill rather than discrete cell-by-cell edits. This consistency suggests that the model is not merely matching static templates, but is transferring controllable execution primitives and organizing them into a stable policy under a new task structure. On O-11, we sometimes observe behaviors resembling "understand → act → adjust". In addition to applying the intended two-step transforma-

tion to the queried shape (first change color, then move it), VBVR-Wan2.2 may modify intermediate elements (e.g., shifting a misaligned reference shape toward the arrow cue; arrows are manually overlaid for visualization), effectively *rationalizing* an internally assumed transformation narrative. While such interventions may conflict with the ground-truth reasoning trace and still yield imperfect final answers, they provide a qualitative signal that the model is maintaining scene-level coherence and executing multi-step plans rather than producing one-shot, uncontrolled scene rewrites.

**Limitations and failure modes (VBVR-Wan2.2).** Despite improved scene controllability, several challenging regimes remain. We observe *process unfaithfulness* in tasks with explicit procedural ground truth. On O-21 (*construction blueprint*), the gold procedure scans candidate pieces one-by-one, previews each candidate at the gap, marks incorrect candidates with a cross, and stops when the correct candidate is found and placed. The generated video can mimic a plausible-looking trial-and-error process without faithfully reflecting the true decision mechanism ("correct answer, wrong method"), highlighting the need for stronger process-level supervision and evaluation. Finally, long-horizon control can break down in interactive tasks. On G-47, compared to the base model that may move doors/keys directly, VBVR-Wan2.2 better distinguishes the agent from scene entities and exhibits the correct high-level subgoal structure (fetch key → reach door). However, it can still suffer from control failures such as agent duplication/flickering when traversing a coherent path, indicating that maintaining identity and stable dynamics over long horizons remains an open problem.

In summary, these qualitative insights reinforce a fundamental shift in evaluating video intelligence: **controllability is the bedrock of verifiable reasoning.** Our results demonstrate that VBVR training moves beyond generic video synthesis, instilling a 'controllability-first' execution logic that generalizes even to novel, out-of-domain task structures. While VBVR-Wan2.2 shows a nascent ability to coordinate multi-step strategies, the remaining gap in process faithfulness and identity stability highlights the next frontier. Achieving true video reasoning will require not just larger scale, but a move toward models that can maintain causal and physical constraints over extended temporal horizons.

*Figure 4.* **Qualitative overview on held-out OOD task families.** Panel A presents same-task, same-sample comparisons between VBVR-Wan2.2 and Sora 2 on three controllable-execution tasks: O-5 (delete the marked symbol with minimal unintended changes), O-6 (apply a 2D geometric rotation under the target cue), and O-30 (rearrange a bookshelf by moving an object into the designated slot), with checkmarks/crosses indicating task success/failure. Panel B shows VBVR-Wan2.2-only emergent behaviors on O-49 (complete a symmetric pattern with a consistent self-chosen policy) and O-11 ("rationalizing": modifying intermediate elements to fit an internally assumed transformation narrative). Panel C reports honest boundaries of VBVR-Wan2.2 on G-47 (long-horizon key–door navigation, with possible agent duplication/flickering) and O-21 (blueprint gap filling, where the video can be correct yet procedurally unfaithful).

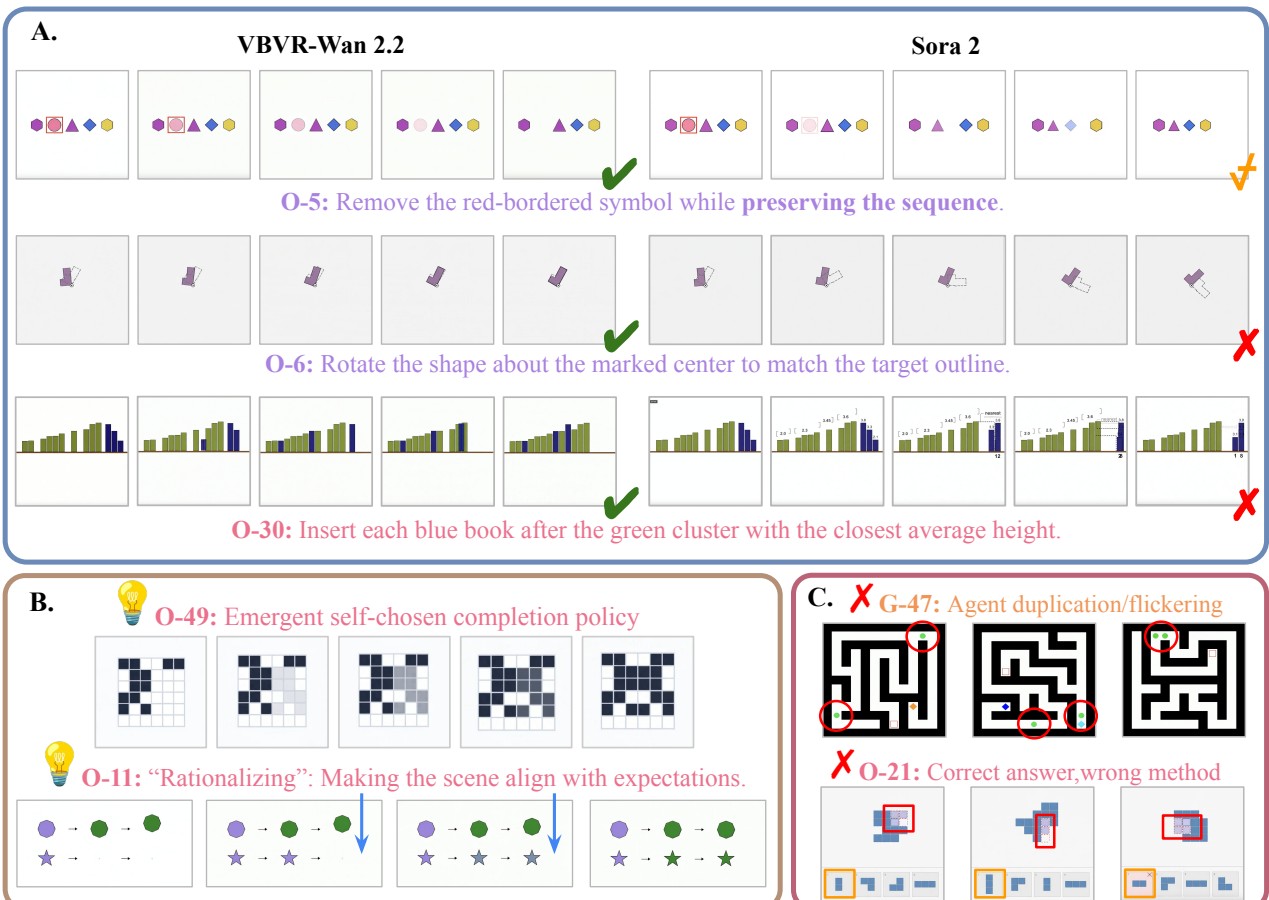

## 5.4. General Performance on VBench++

*Table 4.* Key VBench-I2V metrics for VBVR-Wan2.2 vs. the Wan2.2-I2V-A14B base. Full results across all twelve VBench metrics are in Table 8.

| Model | Total Score | Cam. Motion | Subj. Consist. | Bkgd. Consist. | Dyn. Degree |
|---|---|---|---|---|---|
| Wan2.2-I2V-A14B | 0.8816 | 0.5444 | 0.9752 | 0.9903 | 0.5285 |
| VBVR-Wan2.2 | **0.8835** | **0.6592** | **0.9804** | **0.9921** | 0.4106 |

To evaluate the performance of VBVR-Wan2.2 in real-world video generation scenarios, we benchmarked it against the Wan2.2-I2V-A14B base model using the VBench-I2V suite. As shown in Table 4 (full breakdown across all twelve metrics in Table 8), after LoRA training on VBVR-Dataset, the model maintains a high level of performance across all core metrics, demonstrating that VBVR-Wan2.2 does not undermine the fundamental generative capabilities of the base model. Notably, we observed a significant increase in Video-Text Camera Motion Consistency (rising from 0.5444

to 0.6592), accompanied by a decrease in Dynamic Degree. These quantitative results align closely with our qualitative findings. That is, the model exhibits a more precise understanding of motion dynamics, effectively discerning which regions require temporal change and which should remain preserved. This balance results in videos that are both more stable and better aligned with the provided motion prompts.

## 6. Conclusion

We present VBVR-Dataset, the first large-scale training dataset for video reasoning, and VBVR-Bench, a verifiable, reproducible evaluation toolkit. Our scaling studies show early emergent generalization but a persistent gap that data scaling alone cannot close. Future work could extend evaluation to real-life and 3D embodied domains and pursue architectural advances beyond data scaling. We hope VBVR lays a foundation for generalizable video reasoning.

## Acknowledgements

We thank Amazon Web Services for supporting this work through the AWS Trainium for Research program ([https://aws.amazon.com/ai/machine-learning/trainium/research/](https://aws.amazon.com/ai/machine-learning/trainium/research/)). The compute and storage resources provided through this program were essential to building VBVR at the scale described in this paper.

## Impact Statement

This paper presents work whose goal is to advance the field of machine learning by enabling systematic investigation of video reasoning as a potential new paradigm for building generalist vision foundation models. While training such models may require substantial computational resources that are currently accessible primarily to large institutions, the public release of datasets and evaluation tools aims to lower barriers to entry for research in this area and support transparent, reproducible progress. Overall, the societal implications of this work are consistent with those commonly associated with advances in large-scale machine learning.

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

# A. Details of Cognitive Taxonomy

This appendix provides detailed cognitive science grounding for the five capability dimensions that organize VBVR's task taxonomy (Section 3.1).

**Motivation.**  Evaluating whether video generation models possess genuine reasoning capabilities, rather than superficial pattern matching, requires tasks that isolate specific cognitive operations. We therefore adopt a taxonomy grounded in empirical cognitive and developmental psychology, where each dimension corresponds to a well-characterized family of abilities with known neural substrates and established behavioral paradigms.

The question of how cognitive abilities decompose has a long history. Aristotle organized cognition as a hierarchy of faculties ascending from perception (*aisthêsis*) through imagination (*phantasia*) to understanding (*noûs*) (Aristotle, 1984b;a); Kant argued the mind actively structures experience through *a priori* forms of intuition and categories of understanding (Kant, 1781). Modern computational cognitive architectures formalize cognitive decomposition within information-processing frameworks (Anderson, 2007; Laird et al., 1987; Hélie & Sun, 2010; Langley et al., 2009), and converging neuroscience evidence confirms that the brain is organized into functionally specialized yet richly interconnected modules (Kanwisher, 2010; Sporns et al., 2005; Bertolero et al., 2015).

Our taxonomy does not claim isomorphism with any single theoretical framework. Rather, it identifies five operationally distinct capability dimensions, each with clear empirical precedent and testable behavioral signatures, that collectively span the space of reasoning demands in our task suite: **perception**, **transformation**, **spatiality**, **abstraction**, and **knowledge**.

**Why procedurally generated, abstract stimuli?**  VBVR uses visually simplified, procedurally generated stimuli rather than naturalistic video. This is a deliberate design choice, following the same rationale as ARC (Chollet, 2019) and CLEVR (Johnson et al., 2017): abstract stimuli isolate spatiotemporal reasoning from confounds that pervade naturalistic benchmarks—texture bias, lighting and camera variation, scene-level memorization from web-scale pretraining, and the difficulty of establishing unambiguous ground-truth reasoning traces. In VBVR, every task instance has a deterministic, verifiable solution, and visual simplicity ensures that failures reflect reasoning limitations rather than perceptual noise.

## A.1. Perception

Perception is the extraction of structured representations from sensory input. Helmholtz characterized this as "unconscious inference" (von Helmholtz, 1867; Helmholtz, 1924), and Marr's (1982) tri-level framework—computational theory, algorithm, implementation—remains the organizing paradigm connecting biological and artificial vision. The visual system exemplifies perception's hierarchical architecture. V1 neurons function as oriented edge detectors organized into columnar structures reflecting natural image statistics (Hubel & Wiesel, 1962; 1968; Bonhoeffer & Grinvald, 1991; Hubel & Wiesel, 1977; Olshausen & Field, 1996). Beyond V1, the ventral and dorsal streams achieve progressively invariant object recognition (Ungerleider & Mishkin, 1982; Goodale & Milner, 1992; DiCarlo et al., 2012; Tanaka, 1996), with category-selective regions like the fusiform face area emerging in inferotemporal cortex (Kanwisher et al., 1997; Tsao et al., 2006; Quian Quiroga et al., 2023), where cells collectively map a low-dimensional "object space" (Bao et al., 2020). Predictive processing accounts propose that the brain generates top-down predictions about sensory input and computes prediction errors, with representational geometry shaped by environmental regularities (Rao & Ballard, 1999; Friston, 2010; Keller & Mrsic-Flogel, 2018; Greco et al., 2024; Kersten et al., 2004; Ma et al., 2006). Multimodal integration binds signals across modalities through causal inference in regions like superior temporal cortex (Ghazanfar & Schroeder, 2006; Stein & Stanford, 2008; Calvert, 2001; Körding et al., 2007; Shams & Beierholm, 2010). Deep CNNs exhibit cortex-aligned representations, though they lack robust higher-level processing found in biological systems (Yamins et al., 2014; Kriegeskorte, 2015; Kazemian et al., 2025; Baker et al., 2018; Geirhos et al., 2021). Gibson's ecological approach reminds us that perception ultimately serves action through affordances (Gibson, 2014). In VBVR, perception tasks—edge detection, shape discrimination, figure–ground segmentation, color matching—target the foundational operations that any visual reasoning system must perform before higher-level inference can proceed.

## A.2. Transformation

Transformation refers to the cognitive faculty that manipulates, recombines, and synthesizes mental representations. Shepard and Metzler's (1971) classic demonstration that mental rotation operates analogically on quasi-spatial representations—with reaction times increasing linearly with angular disparity—established transformation as a distinct cognitive operation,

separable from both perception and abstract reasoning. Kosslyn's (1994) imagery theory proposes that mental images are constructed in a "visual buffer" for active manipulation, while Treisman's (1980) feature integration theory shows that focused attention binds preattentively registered features into coherent objects, a process that breaks down with parietal lesions. Neuroimaging localizes transformational operations to the posterior parietal cortex, particularly the intraparietal sulcus (Zacks, 2008; Papadopoulos et al., 2018), and cognitive architectures like ACT-R formalize this through an imaginal module with parietal correlates (Anderson, 2007). In VBVR, transformation tasks—mental rotation, reflection, scaling, and compound operations combining multiple transformations—test whether video models can execute systematic manipulations of represented objects while preserving identity and spatial coherence across frames.

### A.3. Spatiality

Spatiality—the representation of places and their geometric relationships—is one of the earliest-developing and most phylogenetically conserved cognitive capacities. Developmental psychology identifies it as a core knowledge system: infants possess domain-specific machinery for representing places and geometric relationships, with deep roots observable across species and cultures (Spelke & Kinzler, 2007; Spelke, 2000). Tolman (1948) first proposed that animals construct internal "cognitive maps" enabling flexible navigation beyond simple stimulus-response associations. The neural substrates were subsequently revealed: *place cells* in the hippocampus fire at specific locations (O'Keefe & Dostrovsky, 1971), while *grid cells* in the entorhinal cortex provide metric coordinates through hexagonal firing patterns (Hafting et al., 2005)—discoveries recognized by the 2014 Nobel Prize as "a positioning system in the brain" (Moser & Moser, 2014; Nobel Prize Committee, 2014). The posterior parietal cortex complements this system by organizing sensory coordinates into motor-relevant reference frames (Colby & Goldberg, 1999; Andersen et al., 1997). Computational models inspired by grid cells have been proposed, but human-level spatial reasoning remains challenging (Banino et al., 2018; Whittington et al., 2020; Behrens et al., 2018). In VBVR, spatiality tasks—grid navigation, path planning, spatial layout comprehension—probe whether video models maintain coherent geometric representations and can execute spatially grounded plans over extended temporal horizons.

### A.4. Abstraction

Abstraction is the extraction of generalizable patterns and rules from particular instances—the capacity that allows an agent to recognize that two superficially different situations share the same underlying structure. Carey's (2009) account of conceptual change shows that abstract concepts emerge through Quinian bootstrapping, producing representational systems with greater expressive power than core cognition, with concepts possessing causally deep "cores" (Carey, 2011; Keil, 1989). Gentner's (1983) structure-mapping theory explains how analogical reasoning maps relational patterns across domains, with systematicity determining preferred mappings and enabling children to extract relational abstractions (Gentner & Hoyos, 2017; Holyoak, 2012; Holyoak et al., 2001). Neuroscience reveals abstraction's hierarchical implementation: the lateral PFC exhibits a rostral-to-caudal gradient whereby anterior regions process increasingly abstract information, with rostral PFC serving as a "gateway" for internally generated thought (Badre & Nee, 2018; Christoff et al., 2009; Burgess et al., 2007; Ramnani & Owen, 2004). Memory systems contribute through schema-guided consolidation, with hippocampus and vmPFC maintaining abstract prototype representations that support generalization (Bowman & Zeithamova, 2018; Gilboa & Marlatte, 2017; Tse et al., 2011; McClelland et al., 2020). Hierarchical abstraction is also a central principle in deep learning's layered representations (LeCun et al., 2015; Bengio et al., 2013) and cognitive architectures' multiple processing levels (Anderson, 2007; Laird et al., 1987; Tomov et al., 2020). In VBVR, abstraction tasks—Raven's Progressive Matrices, sequence completion, pattern generalization, rule-based sorting—test whether video models can extract relational structure that transfers beyond surface features.

### A.5. Knowledge

The Knowledge dimension in VBVR encompasses domain-specific representational systems that structure reasoning about the physical and symbolic world. This category is grounded primarily in Spelke's (2007) core knowledge framework and related work on intuitive physics.

**Core knowledge and intuitive physics.** Developmental psychology demonstrates that human infants possess, from the earliest months of life, domain-specific representational systems that structure cognition prior to and independently of explicit instruction (Spelke & Kinzler, 2007; Spelke, 2000). These operate according to principles that go well beyond what associative learning from perceptual statistics could deliver. Baillargeon (1985) showed that 5-month-old infants already

expect objects to persist when occluded and solids not to interpenetrate—principles of *continuity* and *solidity* that constitute core intuitive physics. Neuroimaging reveals that physical prediction engages frontal and parietal cortices overlapping with action planning, suggesting a neural "physics engine" that runs probabilistic simulations (Fischer et al., 2016; Battaglia et al., 2013). VBVR's physics-related knowledge tasks (bouncing, gravity, fluid dynamics, reflection, refraction) directly probe these intuitive-physics capacities.

**Learned symbolic knowledge.**    Beyond innate core systems, humans accumulate explicit knowledge through experience— what Carey (2009) terms Quinian bootstrapping, the construction of new representational resources from the combination and enrichment of core systems. VBVR also includes tasks that require learned symbolic knowledge: numerical comparison, clock reading, character recognition, and chart comprehension. These complement the intuitive-physics tasks by testing whether models can leverage structured symbolic representations, not only sensorimotor regularities.

The grouping of intuitive physics and symbolic knowledge under a single dimension reflects a deliberate design choice: both involve applying stored domain-specific knowledge to interpret and predict outcomes, as opposed to the domain-general pattern extraction tested by Abstraction or the spatial-geometric reasoning tested by Spatiality.

Figure 5 summarizes the distribution of the 150 released tasks across these five dimensions, and Table 5 provides the complete task-level taxonomy.

*Figure 5.* Distribution of 150 visual reasoning tasks across five cognitive dimensions in the VBVR-Dataset.

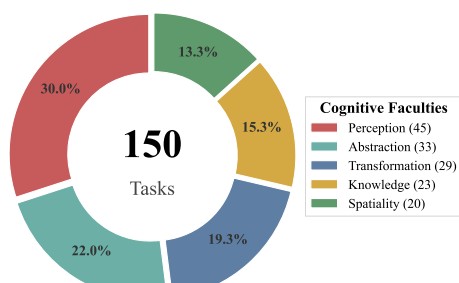

*Table 5.* Complete Cognitive Taxonomy of VBVR-Dataset (150 Visual Reasoning Tasks)

| Task ID | Cognitive Category | Description |
| --- | --- | --- |
| **ABSTRACTION (33)** | | |
| G-7 | Abstraction | Return objects to correct bin by category |
| G-26 | Abstraction | Maintain object identity across different objects |
| G-29 | Abstraction | Find extreme values in chart (with data labels) |
| G-37 | Abstraction | Identify random symmetry patterns |
| G-38 | Abstraction | Identify shape symmetry patterns |
| G-41 | Abstraction | Grid highest cost path finding |
| G-44 | Abstraction | Breadth-first search traversal |
| G-49 | Abstraction | Complete missing contour segments |
| G-51 | Abstraction | Predict next color in sequence |
| G-131 | Abstraction | Select next figure in increasing size sequence |
| G-133 | Abstraction | Select next figure in decreasing size sequence |
| G-134 | Abstraction | Select next figure in large-small alternating sequence |
| G-135 | Abstraction | Select next figure in small-large alternating sequence |
| G-193 | Abstraction | Draw next sized shape in pattern |
| O-7 | Abstraction | Shape color change operations |
| O-8 | Abstraction | Shape rotation operations |
| O-9 | Abstraction | Shape scaling operations |
| O-10 | Abstraction | Shape outline fill operations |
| O-11 | Abstraction | Shape color then move (compound operation) |
| O-12 | Abstraction | Shape color then scale (compound operation) |
| O-13 | Abstraction | Shape outline then move (compound operation) |
| O-14 | Abstraction | Shape scale then outline (compound operation) |
| O-21 | Abstraction | Construction blueprint interpretation |
| O-29 | Abstraction | Ball color clustering and merging |
| O-30 | Abstraction | Bookshelf organization task |
| O-37 | Abstraction | Light sequence pattern recognition |
| O-43 | Abstraction | Object subtraction (quantity reasoning) |

*Table 5 — continued from previous page*

| Task ID | Cognitive Category | Description |
|---------|-------------------|-------------|
| O-45 | Abstraction | Sequence completion task |
| O-47 | Abstraction | Sliding puzzle solving |
| O-49 | Abstraction | Symmetry completion task |
| O-54 | Abstraction | Control panel symbol manipulation |
| O-56 | Abstraction | Raven's Progressive Matrices reasoning |
| O-66 | Abstraction | Animal color sorting task |
| **KNOWLEDGE (23)** | | |
| G-27 | Knowledge | Read chart data and semantic comprehension |
| G-30 | Knowledge | Find extreme values in chart (without data labels) |
| G-35 | Knowledge | Hit target after bounce |
| G-48 | Knowledge | Multiple bounces prediction |
| G-160 | Knowledge | Circle largest numerical value |
| G-162 | Knowledge | Locate twelve o'clock arrows |
| G-163 | Knowledge | Identify digits 1 and 9 |
| G-200 | Knowledge | Circle maximum value in set |
| G-217 | Knowledge | Circle central dot |
| G-247 | Knowledge | Identify Chinese character |
| G-273 | Knowledge | High density liquid behavior |
| O-3 | Knowledge | Symbol reordering operations |
| O-15 | Knowledge | Ball bounces at given time |
| O-18 | Knowledge | Glass refraction |
| O-19 | Knowledge | Mirror reflection |
| O-23 | Knowledge | Domino chain branch path prediction |
| O-24 | Knowledge | Domino chain gap analysis |
| O-34 | Knowledge | Dot-to-dot connection task |
| O-52 | Knowledge | Traffic light state reasoning |
| O-53 | Knowledge | Clock reading and time reasoning |
| O-62 | Knowledge | Gravity physics simulation |
| O-75 | Knowledge | Communicating vessels (fluid dynamics) |
| O-87 | Knowledge | Fluid diffusion reasoning |
| **PERCEPTION (45)** | | |
| G-3 | Perception | Stable sort objects maintaining order |
| G-4 | Perception | Identify and distinguish different objects |
| G-5 | Perception | Multi-object placement to specified positions |
| G-9 | Perception | Identify objects in specified region |
| G-19 | Perception | Sort objects by specified rule |
| G-22 | Perception | Attention shift to same object |
| G-39 | Perception | Attention shift to different object |
| G-43 | Perception | Understand scene structure and spatial layout |
| G-54 | Perception | Connecting matching colors |
| G-132 | Perception | Find fragment for gap filling |
| G-136 | Perception | Locate point in overlapping area |
| G-137 | Perception | Identify figure in overlapping area |
| G-138 | Perception | Spot unique non-repeated color |
| G-141 | Perception | Identify polygon with most sides |
| G-143 | Perception | Select box with most dots |
| G-146 | Perception | Circle all squares from mixed shapes |
| G-147 | Perception | Identify unique figure in uniform set |
| G-158 | Perception | Identify all hollow points |
| G-161 | Perception | Mark second largest shape |
| G-165 | Perception | Mark tangent point after motion |
| G-166 | Perception | Highlight horizontal lines |
| G-167 | Perception | Select longest polygon side |
| G-168 | Perception | Identify rectangle nearest to square |
| G-169 | Perception | Locate intersection of line segments |
| G-174 | Perception | Arrange circles by circumference |
| G-189 | Perception | Draw midpoint perpendicular line |
| G-195 | Perception | Select nearest 2:1 rectangle |
| G-198 | Perception | Mark right-angled triangles |
| G-199 | Perception | Locate line intersections |
| G-202 | Perception | Mark wave peaks |
| G-206 | Perception | Identify pentagons |
| G-212 | Perception | Find incorrect arrow direction |
| G-218 | Perception | Identify largest angle in triangle |
| G-222 | Perception | Mark tangent point of circles |
| G-223 | Perception | Highlight horizontal lines |
| G-248 | Perception | Mark asymmetrical shape |
| G-250 | Perception | Color triple intersection red |
| O-1 | Perception | Color mixing (additive) |
| O-2 | Perception | Pigment color mixing (subtractive) |
| O-16 | Perception | Color addition operations |
| O-17 | Perception | Color subtraction operations |
| O-31 | Perception | Ball eating mechanics |
| O-33 | Perception | Counting objects accurately |
| O-38 | Perception | Identify majority color in set |

| Task ID | Cognitive Category | Description |
|---|---|---|
| *Table 5 — continued from previous page* | | |
| O-65 | Perception | Animal size sorting |
| **SPATIALITY (20)** | | |
| G-12 | Spatiality | Grid navigation to obtain reward |
| G-13 | Spatiality | Grid number sequence navigation |
| G-14 | Spatiality | Grid color sequence navigation |
| G-15 | Spatiality | Grid navigation avoiding obstacles |
| G-16 | Spatiality | Grid navigation going through blocks |
| G-17 | Spatiality | Grid navigation avoiding red blocks |
| G-18 | Spatiality | Grid shortest path finding |
| G-31 | Spatiality | Directed graph navigation |
| G-32 | Spatiality | Undirected graph navigation |
| G-33 | Spatiality | Visual Jenga game |
| G-45 | Spatiality | Key-door matching puzzle |
| G-46 | Spatiality | Find keys and open doors |
| G-47 | Spatiality | Multiple keys for one door puzzle |
| G-140 | Spatiality | Locate topmost unobscured figure |
| G-219 | Spatiality | Select leftmost shape |
| G-221 | Spatiality | Outline innermost square |
| O-25 | Spatiality | LEGO construction assembly |
| O-39 | Spatiality | Maze navigation and solving |
| O-55 | Spatiality | Rotation operations |
| O-83 | Spatiality | Planar warp verification |
| **TRANSFORMATION (29)** | | |
| G-1 | Transformation | Predict object trajectory |
| G-2 | Transformation | Reorder objects by rule |
| G-6 | Transformation | Resize object |
| G-8 | Transformation | Track object movement |
| G-11 | Transformation | Handle object reappearance after disappearance |
| G-21 | Transformation | Multiple occlusions (vertical) |
| G-24 | Transformation | Separate objects (no rotation) |
| G-25 | Transformation | Separate rotating objects |
| G-34 | Transformation | Object packing optimization |
| G-36 | Transformation | Multiple occlusions (horizontal) |
| G-40 | Transformation | Combined rotating objects |
| G-50 | Transformation | Suppress spurious edges |
| G-194 | Transformation | Construct concentric ring |
| G-240 | Transformation | Add borders to unbordered shapes |
| O-4 | Transformation | Symbol substitution operations |
| O-5 | Transformation | Symbol deletion operations |
| O-6 | Transformation | 2D geometric transformation |
| O-22 | Transformation | Construction stack (gravity balance) |
| O-27 | Transformation | Move 2 objects to 2 targets |
| O-32 | Transformation | Rolling ball physics |
| O-36 | Transformation | Grid shift operations |
| O-44 | Transformation | Rotation puzzle |
| O-46 | Transformation | Shape sorter classification |
| O-58 | Transformation | Symbol delete operations |
| O-59 | Transformation | Symbol insert operations |
| O-60 | Transformation | Symbol substitute operations |
| O-61 | Transformation | Symbol edit operations |
| O-64 | Transformation | Animal matching task |
| O-85 | Transformation | 2D object rotation |

# B. Data Curation

This appendix provides a comprehensive account of the data curation pipeline that underlies the VBVR-Bench. We detail the five-stage process from task design to benchmark construction, the quality assurance mechanisms, and the infrastructure that enables large-scale data generation.

## B.1. Overview of Data Curation Pipeline

*Figure 6.* Task designs grounded in cognitive taxonomy are implemented as parameterized generators, then executed at scale via distributed Lambda workers writing to centralized S3 storage.

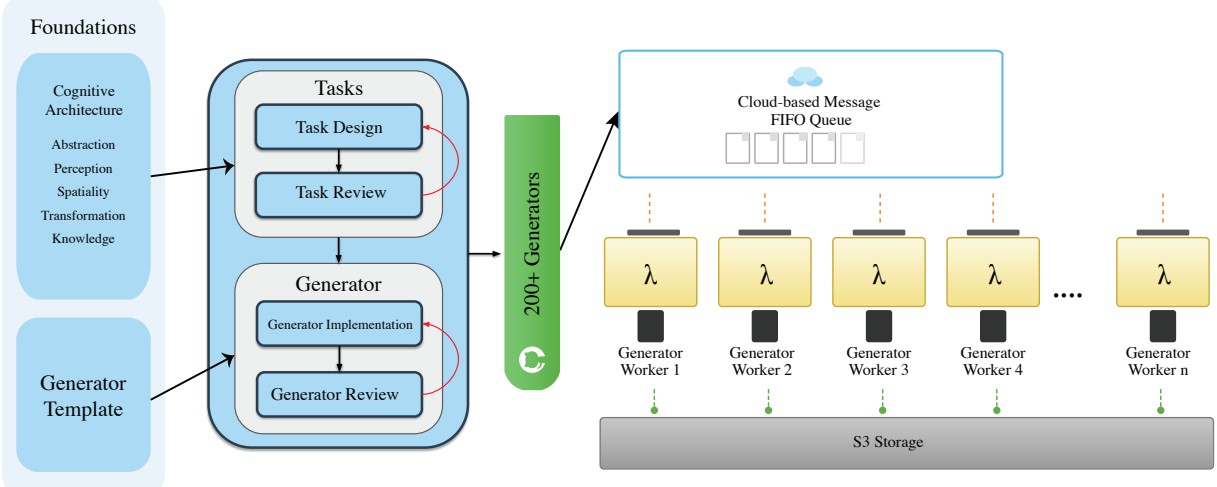

The VBVR-Bench is built through a systematic five-stage pipeline (Figure 6) that transforms cognitive task concepts into a standardized evaluation framework.

**Pipeline Flow**. The data curation pipeline consists of five sequential stages. **Stage 1: Task Design** We begin by designing cognitive reasoning tasks grounded in cognitive science principles. Starting from over 300 task candidates, we employ a dual-review process to select tasks that meet six quality criteria. This stage produces two distinct task sets: 100 training tasks and 100 testing tasks with an overlapping of 50 tasks. **Stage 2: Task Implementation** Each training task is implemented as a parameterized generator capable of producing diverse samples. These generators form the our data spring, VBVR-DataFactory, a set of modular synthesis engines. All generators inherit from a standardized `BaseGenerator` template and are managed as independent repositories within a GitHub Organization. Implementation undergoes rigorous quality control by dedicated reviewers who verify scalability, code quality, and adherence to file specifications. **Stage 3: Large-Scale Distributed Generation**. The VBVR-DataFactory infrastructure orchestrates parallel generation of one million training samples (10,000 per task) stored privately on AWS S3. Simultaneously, we generate 500 fresh test cases (5 per task) from the Test task set. Each sample comprises an initial frame, task prompt, ground-truth solution, and reference video (Figure 7). **Stage 4: Model Evaluation**. We employ VBVR-EvalKit to evaluate eight state-of-the-art image-to-video (I2V) baseline models, along with our fine-tuned VBVR-Wan2.2, on the 500 test cases, producing 4,500 generated videos. Human annotators assess each video across three reasoning-specific dimensions: Task Completion, Reasoning Logic, and Visual Quality, each scored on a 1-5 scale. **Stage 5: Benchmark Construction and Model Training**. The evaluation data is integrated into a standardized benchmark dataset. We train the VBVR-Wan on the one million training samples, demonstrating that explicit reasoning supervision significantly improves performance on reasoning-intensive tasks.

### B.1.1. KEY STATISTICS

Table 6 summarizes the scale and coverage of the VBVR benchmark:

*Table 6.* Data Construction Statistics

| Component | Scale | Description |
|---|---|---|
| Task Candidates | 300+ | Initial task pool |
| Training Tasks | 100 | Generate training data |
| Test Tasks | 100 | 50 In-Domain + 50 Out-of-Distribution |
| Training Samples | 1,000,000 | 100 tasks × 10,000 samples per task |
| Test Cases | 7,500 | 150 tasks × 50 samples per task |
| Cognitive Categories | 5 | Systematic coverage |
| Evaluated Models | 9 | 8 SOTA I2V baselines (4 open-source, 4 commercial) + our fine-tuned VBVR-Wan2.2 |
| Generated Videos | 4,500 | 9 models × 500 cases |
| Evaluation Records | 4,500 | Task Completion/Reasoning Logic/Visual Quality annotations |
| Development Team | 75 | 53 OSS coders + 7 full-time employees + 5 QC reviewers |
| Code Repositories | 300+ | Github Organization |

### B.1.2. DESIGN PRINCIPLES

The pipeline is guided by the following core design principles: **1. Dual Generalization Testing**. The In-Domain/OOD split enables evaluation of both in-task generalization (seen task types with unseen samples) and cross-task generalization (unseen task types). **2. Reasoning-First Paradigm**. Unlike benchmarks that primarily assess generation quality, VBVR emphasizes reasoning correctness through TC, RL, and VQ metrics that capture task completion, logical consistency, and visual fidelity. **3. Industrial-Scale Quality Assurance** We employ a dual-review mechanism: peer review for task design (by five task designers) and dedicated quality control for implementation (by more than two specialized reviewers). Six quality criteria ensure tasks are well-specified, visually clear, and scalable. **4. Extensible Infrastructure**. The modular generator architecture and comprehensive evaluation toolkit facilitate easy extension with new tasks and models. **5. Cognitive Inspiration**. Our task design began with a loose cognitive taxonomy inspired by cognitive science literature. However, rather than rigidly adhering to a fixed theoretical framework, we adopted a task-driven design philosophy: we prioritized designing intellectually meaningful reasoning tasks first, then organized them into our categories. **6. Diversity Focus**. This approach contrasts with top-down frameworks that first define strict categories, then fill them with tasks. Our bottom-up approach ensured that each task possesses genuine reasoning value, and maximizing our diversity, rather than forcing tasks to fit narrow pre-determined boxes.

### B.1.3. TASK SELECTION AND FILTERING

**Initial Pool**   We designed over 300 task candidates through iterative brainstorming sessions. Each designer independently proposed tasks, which were then collectively discussed and refined.

**Selection Criteria (Six Standards)**   Tasks were evaluated against six criteria during peer review:

1. **Information Sufficiency**: The first frame must contain all information necessary for reasoning, without requiring external context.

    ✓ Pass Example: G-15 (Grid Avoid Obstacles) — the first frame shows start point, end point, and all obstacles.

    ✗ Fail Example: Tasks where the goal object is initially occluded or requires additional verbal explanation.

2. **Deterministic Solution**: Tasks should have a clear, unique solution (or explicitly defined criteria for multiple valid solutions, e.g., "shortest path").

    ✓ Pass Example: G-18 (Grid Shortest Path) — the optimal path is unambiguous.

    ✗ Fail Example: Open-ended tasks with ambiguous success conditions.

3. **Video-Based Reasoning**: Tasks must be suitable for video generation models, encompassing diverse reasoning types (temporal dynamics, static recognition, logical inference).

    ✓ Pass Example: O-23 (Domino Chain) — requires temporal causality reasoning.

    ! Note: Unlike some benchmarks, we do NOT exclude static recognition tasks; we include both dynamic and static reasoning.

4. **Visual Clarity**: Objects must be distinguishable, text/numbers legible, and layouts uncluttered to avoid perceptual ambiguity.

5. **Parametric Diversity (Scalability)**: The parameter space must be large enough to generate 10,000 non-trivial, distinct samples per task.

6. **Technical Feasibility**: Tasks must be implementable with our rendering pipeline (PIL-based), output standard file formats, and avoid edge cases (e.g., initial occlusions, boundary overflow).

**Rejection Cases**   Here are some of the tasks were excluded:

- **A. Multi-Step Logic Chains**

   – Example: Logic Gate Circuits
   – Reason: Reasoning chains too long to clearly visualize in video format.

- **B. Complex Physics Requiring Precise Numerical Solutions**

   – Example: Elastic Collisions with friction coefficients
   – Reason: Continuous parameters yield no deterministic solution (tiny parameter variations lead to drastically different trajectories); I2V models cannot precisely simulate physics.

- **C. Excessively Difficult Tasks**

   – Example: Sudoku
   – Reason: High difficulty even for humans; unsuitable for video format (better suited for static reasoning); evaluation criteria too strict (any single cell error constitutes failure).

### B.1.4. PEER REVIEW PROCESS

All task designs underwent a structured peer review by the five task designers, with each designer evaluating others' proposals using a standardized checklist derived from the six quality criteria; tasks typically progressed through 2–3 rounds of iterative refinement based on this feedback, during which common issues were identified, including **ambiguous success conditions**, **insufficient visual clarity**, **parameter spaces too small to support 10k samples**, and **unintentional edge cases** (e.g., objects overlapping at initialization). A task advanced to implementation (Stage 2) only after receiving consensus approval from at least three reviewers. This process ensured that every task in VBVR meets rigorous cognitive and technical standards, forming a solid foundation for the subsequent implementation and evaluation stages.

### B.2. Details of Task Implementation

This section details the implementation of the 200 tasks as parameterized generators. We describe the standardized architecture, code management infrastructure, parameterization strategies, visual rendering pipeline, and dedicated quality control procedures.

### B.2.1. GENERATOR ARCHITECTURE AND STANDARDIZATION

**BaseGenerator Template**   All task generators inherit from a standardized `BaseGenerator` abstract base class, ensuring consistent interfaces and output formats. The template defines:

1. **Required Methods**:
   - `__init__(self, seed, params)`: Initialize generator with random seed and task-specific parameters.
   - `generate_sample(self) -> Sample`: Generate a single task instance.
   - `validate_sample(self, sample) -> bool`: Verify sample meets quality criteria.
   - `save_sample(self, sample, output_dir)`: Save sample to standardized file structure.

2. **Standard Output Format**: Each sample is saved as a directory containing:
   - `first_frame.png`: Initial state image (required)
   - `prompt.txt`: Task instruction in natural language (required)

- `final_frame.png` or `goal.txt`: Target state or goal description
- `ground_truth.mp4`: Reference solution video demonstrating the correct reasoning process

3. **Configuration Management**: Generators expose configurable parameters through a `config.yaml` file specifying:

   - Parameter ranges (e.g., grid size: 5-10, number of objects: 3-8)
   - Difficulty levels (easy/medium/hard)
   - Visual settings (resolution: 512×512, frame rate: 24 fps)

**Design Rationale**. The `BaseGenerator` template serves three core purposes: **Consistency**, by providing uniform interfaces that simplify integration with VBVR-DataFactory for batch generation; **Quality Assurance**, through the `validate_sample()` method, which enforces critical checks such as ensuring no object occlusions and that valid solutions exist; and **Extensibility**, enabling new tasks to be added by implementing the abstract methods while leveraging shared utility functions.

### B.2.2. CODE MANAGEMENT INFRASTRUCTURE

All task generators are managed as independent repositories within a GitHub Organization, an architecture that provides clear organizational and operational advantages: each repository follows a strict Naming Convention of `{Type}-{ID}_{task_name}_data-generator` (e.g., `G-3_stable_sort_data-generator` and `O-15_ball_bounces_given_time_data-generator`), where Type denotes G, which are contributed by commercial enterprises, or O, which are contributed by OSS developers, and ID is a unique numeric identifier; this structure enables Independent Versioning, as each task maintains its own commit history, tags, and release cycles, and supports Modular Updates, ensuring that bug fixes or parameter adjustments to one task do not affect others.

All generators share the `core/` directory structure through Git submodules or package dependencies, ensuring consistent utility functions while allowing each task to evolve independently. This design yields several version control benefits: 1. Traceability, as each sample records the exact generator version (git commit hash) used for generation; 2. Reproducibility, enabling researchers to check out specific generator versions to reproduce sample generation; and 3. Collaboration, allowing multiple team members to work on different tasks simultaneously without merge conflicts.

*Figure 7.* This is a typical example of data samples in our dataset. The model receives a prompt and a first image, and is asked to generate a video that solves the prompt. Final image and ground truth videos are provided as references in data samples.

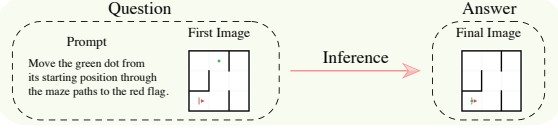

### B.2.3. PARAMETERIZATION STRATEGY FOR DIVERSITY

To generate 10,000 distinct, non-trivial samples per task, we employ systematic parameterization strategies:

**Example 1: G-15 (Grid Avoid Obstacles)**

```
Parameters:
  - grid_size: [5, 6, 7, 8, 9, 10]        # 6 options
  - num_obstacles: [3, 4, 5, ..., 15]     # 13 options
  - start_position: grid-dependent        # ~grid_size² options
  - end_position: grid-dependent          # ~grid_size² options
  - obstacle_layout: random_valid         # ~10^6 variations

Estimated unique samples: > 10^9
Sampling strategy: Constrained random sampling ensuring solvability
```

**Example 2: O-56 (Raven's Progressive Matrices)**

```
Parameters:
  - pattern_type: [size_progression, rotation, color_change,
                   shape_replacement, combination]      # 5 types
  - num_objects: [1, 2, 3, ..., n]                      # n options
  - object_shapes: [circle, square, triangle, ...]      # 8 shapes
  - progression_complexity: [simple, compound]          # 2 levels
  - visual_style: [minimal, decorated]                  # 2 styles

Estimated unique samples: > 10^5
Sampling strategy: Balanced distribution across pattern types
```

**Example 3: G-3 (Stable Sort)**

```
Parameters:
  - num_shape_types: [2, 3, 4]              # 3 options
  - shapes_per_type: [2, 3, 4]              # 3 options
  - color_palette: [vibrant, pastel, ...]   # 5 palettes
  - initial_layout: random_permutation      # n! permutations
  - shape_geometry: [geometric, organic]    # 2 styles

Estimated unique samples: > 10^6
Sampling strategy: Uniform sampling across difficulty levels
```

Diversity is enforced through four complementary mechanisms: stratified sampling, which divides the parameter space into strata (e.g., difficulty levels) and samples proportionally from each; constraint satisfaction, which ensures that all generated samples are valid (e.g., mazes admit solutions and puzzles are solvable); duplicate detection, which tracks parameter combinations via hash functions to prevent exact duplicates; and visual variety, which randomizes visual attributes such as colors, sizes, and positions beyond the core reasoning parameters.

Each generated sample undergoes automatic validation, including a solvability check to verify that a ground-truth solution exists (e.g., via A* search for navigation tasks), a visual clarity check to ensure that objects do not overlap excessively and that text remains legible with a minimum font size, and a parameter bounds check to confirm that all parameters fall within their specified ranges.

### B.2.4. VISUAL RENDERING PIPELINE

All visuals adhere to fixed specifications: a resolution of $512 \times 512$ pixels, 24-bit RGB color depth (8 bits per channel), a frame rate of 24 fps for ground-truth videos, and H.264 as the video codec.

### B.3. Large-Scale Distributed Generation of Training Data

The VBVR-DataFactory component orchestrates the parallel generation of one million training samples using the 300+ generators in our data spring. It provides a complete, production-grade serverless system for cloud infrastructure, generation workflow, data organization, and quality assurance, enabling efficient, reliable, and cost-effective large-scale data production.

All generated samples are stored on Amazon Web Services (AWS) Simple Storage Service (S3). S3 is chosen for its ability to scale to petabyte-level data with automatic capacity management, its high durability through redundant storage across multiple availability zones, and its fine-grained access control through IAM policies. Training data are kept in private buckets with server-side encryption to prevent public exposure and test-set contamination. In addition, S3's tiered storage model makes long-term archival cost-effective.

To efficiently generate one million samples, VBVR-DataFactory employs distributed serverless processing via AWS Lambda. The system distributes tasks across up to 990 concurrent Lambda function instances, each configured with 3 GB memory and a 15-minute execution timeout. Generation tasks are queued in Amazon SQS and automatically trigger Lambda invocations, eliminating the need for manual cluster management. Within each task, samples are produced in configurable batches of 25–100 to balance memory usage and processing efficiency. A typical one-million-sample generation completes in approximately 2–4 hours with the full fleet of 990 concurrent workers, at a total cost of roughly $800–1200 per run (primarily

Lambda compute and S3 storage). Fault tolerance is built in through SQS's automatic retry mechanism with configurable visibility timeouts, a Dead Letter Queue (DLQ) for failed messages after one retry attempt, and CloudWatch metrics for real-time monitoring of task success rates and processing durations.

The end-to-end generation pipeline begins with configuration and planning, where generators, sample counts, random seeds, and output formats are defined. Tasks are submitted to the SQS queue as Pydantic-validated JSON messages, each specifying the generator type, number of samples, starting index for global sample numbering, random seed, and output format (individual files or compressed tar archives). Lambda functions receive these messages, execute the corresponding generator subprocess, validate and rename outputs with zero-padded global indices, upload results to S3 with structured prefixes, and emit success metrics to CloudWatch. Failed tasks automatically move to the DLQ for manual inspection and resubmission after debugging.

On a per-task basis (100 samples, typical batch size), sample generation requires approximately 15 minutes, with generation speed varying significantly by task complexity—simple geometric tasks average 1–3 seconds per sample, while complex graph or physics tasks require 9–15 seconds per sample. File organization and S3 upload add 1–2 minutes depending on file sizes, and Lambda cold starts contribute an additional 5–10 seconds for the first invocation. Each task therefore completes in roughly 15–20 minutes on a Lambda instance, enabling the entire one-million-sample corpus to be produced in 2–4 hours with 990 parallel workers. For efficient access during training, VBVR-DataFactory maintains a hierarchical S3 structure organized by generator and task, with each sample directory containing standardized files (first_frame.png, prompt.txt, and optionally final_frame.png and ground_truth.mp4). This organization allows training pipelines to locate, filter, and stream data efficiently at scale through S3's prefix-based listing and parallel download capabilities.

Quality assurance is integrated throughout generation. Every sample is validated by the generator's internal checks before being saved, ensuring solvability, visual clarity, and file integrity. The system continuously monitors task success rates, processing durations, and samples uploaded through CloudWatch metrics, with alarms configured to detect anomalies such as high failure rates or DLQ accumulation. Sample diversity is enforced through unique per-task random seeds that increment deterministically across batches, preventing duplicate generation. The DLQ captures and preserves failed tasks for post-mortem analysis, enabling systematic debugging of generator issues and infrastructure failures. Across typical production runs, task validation failures occur in fewer than 1 percent of cases, primarily due to edge-case parameter combinations that are caught by Pydantic schema validation before reaching the generator. Infrastructure-related failures (timeouts, out-of-memory errors) occur in approximately 0.1–0.5 percent of tasks and are resolved through DLQ resubmission after adjusting batch sizes or addressing generator bugs.

### B.3.1. AUTOMATED QUALITY CONTROL

Building on the generator-internal `validate_sample()` hook, every sample passes a shared validation pipeline before it is kept: we verify that (i) a valid solution exists, (ii) the rendered frames satisfy task-specific visual and layout rules, and (iii) all sampled parameters stay within each generator's declared ranges. Failed cases are retried automatically, and repeated failures are logged for manual inspection and follow-up. We unpack the three layers of this pipeline below.

**Visual compliance.** Because our scoring is rule-based, the rendered frames must be unambiguous. Validators reject layouts where objects overlap beyond each task's allowed tolerance: the initial frame is expected to carry all information needed for that instance, so heavy overlap would occlude shapes (e.g., in ordering or multi-object layouts) and make boundaries indeterminate for models that only see the first image. When tasks render text or symbolic marks on the canvas, we also enforce minimum font and stroke sizes so that on-frame cues remain legible to video models.

**Boundary constraints.** These are enforced by combining a parameter-bounds pass (each field must stay inside the ranges configured for that task) with validity constraints produced during sampling. For navigation-style tasks we require solvable worlds and verify solvability with the same search-style checks we use in development (e.g., BFS-based pathfinding). Representative generator configurations use bounded discrete ranges such as grid sizes from 5 to 10 (inclusive) and object counts from 3 to 8 for maze-style instantiations, which keeps the sampled space aligned with what the renderer and the evaluator expect.

**Edge cases.** Edge cases are handled in two layers. Task admission already avoids structural pathologies known to break rendering or evaluation (for example boundary overflow off the canvas or goals that are initially fully occluded). When rare runtime faults still occur, the production workflow described above retries, logs persistent failures, and tracks aggregate

validation failure rates so that we can patch generators without manual review of every clip.

All training samples are stored in private S3 buckets with server-side encryption (SSE-S3), versioning disabled to reduce costs, and IAM role-based access control restricted to authorized Lambda functions and training pipelines. Full S3 access logging is enabled for audit trails. The data contain no personally identifiable information, as all samples are fully synthetic and procedurally generated. Together, these design choices demonstrate that, with proper serverless engineering, million-scale, high-quality data generation is achievable, reliable, and cost-effective. The resulting one million samples form the training foundation for VBVR-Wan and future video reasoning systems.

## C. VBVR-Bench Details

### C.1. Model Inference Infrastructure

VBVR-Bench is an end-to-end evaluation framework that integrates large-scale model inference (or API-based invocation for proprietary models) with a rule-based evaluation engine over a standardized 100-task video reasoning benchmark. At the model level, VBVR-Bench provides a unified `VideoGenerator` abstraction that encapsulates 29 video generation systems, including closed-source models such as Google Veo, OpenAI Sora, and Runway Gen-4, as well as open-source and research models such as CogVideoX, Wan2.2, LTX-2, and HunyuanVideo. This abstraction ensures that all models are executed through an identical pipeline, eliminating confounding factors introduced by model-specific inference logic. It supports batch execution at scale, incorporates caching mechanisms to avoid redundant generations, and is fully reusable, allowing interrupted evaluations to continue without loss of progress.

#### C.1.1. MODEL SELECTION AND CONFIGURATION

For the VBVR benchmark, we evaluate eight state-of-the-art image-to-video (I2V) models representing diverse architectures and capabilities:

*Table 7.* Evaluated Models

| Model | Developer | Architecture | Parameters | Key Features |
|---|---|---|---|---|
| CogVideoX1.5-5B-I2V | Tsinghua University / Zhipu AI | Transformer-based diffusion model | 5 billion | Strong text-image alignment |
| Wan2.2-I2V-A14B | Wanx AI | Latent diffusion with attention mechanisms | 14 billion | High-fidelity temporal consistency |
| LTX-2 | Lightricks | Efficient transformer architecture | $\sim$3 billion | Fast inference |
| HunyuanVideo-I2V | Tencent Hunyuan | Multi-stage diffusion with spatial-temporal attention | $\sim$7 billion | Strong on complex scenes |
| Veo | Google DeepMind | Proprietary (closed-source) | Unknown (large-scale) | Strong physics understanding |
| Sora | OpenAI | Proprietary (closed-source, rumored diffusion transformer) | Unknown (likely >10B) | Temporal coherence, realistic physics |
| Runway Gen-3 | Runway ML | Proprietary (closed-source) | Unknown | Creative generation, text adherence |
| Kling 2.6 | Kuaishou Technology | Proprietary (closed-source, likely diffusion-based) | Unknown | Fast inference |

The models included in the benchmark are chosen to reflect the breadth and maturity of the current image-to-video landscape. The set intentionally mixes open-source and closed-source systems, ensuring that both academic and industrial approaches are represented. It spans a wide range of architectural paradigms, capturing fundamentally different design philosophies in video generation. The selected models cover a broad scale spectrum, from approximately 3B parameters to 14B+ and beyond, enabling analysis of how reasoning capability correlates with model size. All included systems represent state-of-the-art performance as of late 2025 and early 2026, ensuring that the benchmark reflects the current frontier of the field.

To ensure fair and controlled comparison, all generated videos follow the resolution of their corresponding ground-truth videos, which include both square and rectangular aspect ratios. For open-source models that support custom input resolutions, we specify the resolution to match the ground truth. For closed-source models with fixed resolution constraints (e.g., Sora 2 only supports 1280×720 or 720×1280), we pad the first-frame image with a background color and automatically detect and remove the padding during evaluation. Frame rates range from 15–24 fps depending on model requirements. For API-based models, we adopt the default or provider-recommended generation settings, while for open-source models we use the configurations reported in their original papers. This standardization removes confounding factors arising from resolution or tuning differences, ensuring that observed performance differences reflect reasoning ability rather than generation format.

### C.2. Details of Human Preference Analysis

This section provides the full methodology behind the human preference alignment study summarized in Section 4.2 (the model-level agreement between VBVR-Bench's automatic win ratios and human preference win ratios is visualized in Figure 3).

#### C.2.1. DATA PREPARATION

To mitigate potential biases in pairwise comparisons, we adopt two complementary scoring schemes: **relative scoring** based on pairwise preference judgments, and **absolute scoring** based on independent per-video ratings. In cases where the absolute scores clearly indicate that one video is superior but the corresponding relative annotation contradicts this assessment, the final decision is revised in favor of the absolute judgment. For each sample consisting of a text prompt and an initial image pair $p_i$, we generate videos using nine video generation models $\mathcal{M} = \{M_1, M_2, M_3, M_4, M_5, M_6, M_7, M_8, M_9\}$, producing a set of outputs $G_i = \{V_{i,1}, V_{i,2}, V_{i,3}, V_{i,4}, V_{i,5}, V_{i,6}, V_{i,7}, V_{i,8}, V_{i,9}\}$.

For relative evaluation, all pairwise combinations are constructed within each group, resulting in $\binom{9}{2} = 36$ unique video pairs per sample. For absolute evaluation, the dataset contains 500 distinct prompt–image pairs $p_i$. Each pair is processed by all nine models, yielding a total of $500 \times 9 = 4{,}500$ single-video annotation instances.

Each video is presented together with its corresponding text prompt and task-specific evaluation documents. To reduce annotation noise, every video is independently annotated five times. All samples are randomly shuffled before assigned to annotators.

### C.2.2. HUMAN ANNOTATION

Annotators are between 20 and 35 years old and possess basic domain knowledge relevant to the tasks. All annotators undergo standardized training to ensure consistent interpretation of evaluation criteria.

For relative scoring, annotators are shown two videos generated from the same prompt and are asked to select which one better completes the task and aligns with the given prompt overall, with ties allowed.

For absolute scoring, annotators rate each video along three dimensions using a 5-point Likert scale, where higher scores indicate better performance. **Task Completion (TC)** assessing whether the task goal is achieved; **Reasoning Logic (RL)**, assessing the correctness of the reasoning process; and **Visual Quality (VQ)**, assessing visual clarity, temporal coherence, and rendering fidelity.

### C.2.3. WIN RATIO CALCULATION

From relative human annotations, we compute a win ratio for each model. In each comparison, the preferred model receives a score of 1 and the other 0; in the case of a tie, both receive 0.5. Win ratios are aggregated across all pairwise comparisons for each evaluation split.

From absolute annotations, we average the scores of the TC, RL and VQ dimensions to obtain a final score per sample. These absolute scores are used for cross-verification of pairwise results and for resolving contradictory annotations between the two scoring schemes.

Finally, we compare the win ratios derived from relative human annotations (with cross-verification using absolute scores) with the win ratios computed from VBVR-Bench 's automatic evaluation metrics, and measure the correlation between the two across models.

## C.3. Detailed Analysis Protocols and Additional Results

### C.3.1. RESIDUALIZED CAPABILITY CORRELATION

This section details the computation behind Fig. 9 in the main paper. Our goal is to quantify *capability dependency* between cognitive categories while avoiding trivial correlations induced by overall model strength.

**Category scores.** Let $m$ index models and $c$ index categories (five total). For each model $m$ and category $c$, we compute a category-level score by averaging the per-sample Overall ratings:

$$S_{m,c} = \text{mean}(\text{Overall}) \quad \text{over all evaluated samples with Category} = c \text{ for model } m.$$

**General factor.** We define a model-level general factor as the overall mean score across all samples:

$$G_m = \text{mean}(\text{Overall}) \quad \text{over all evaluated samples for model } m.$$

**Residualization.** For each category $c$, we regress $S_{m,c}$ on $G_m$ across models:

$$S_{m,c} = a_c + b_c G_m + \epsilon_{m,c},$$

and retain the residuals $\epsilon_{m,c}$ as the *strength beyond overall model quality*.

**Capability dependency matrix.** For each pair of categories $(c_1, c_2)$, we compute Pearson correlation across models using residuals:

$$R_{c_1,c_2} = \text{corr}\left(\{\epsilon_{m,c_1}\}_m, \{\epsilon_{m,c_2}\}_m\right).$$

The resulting $5 \times 5$ matrix $R$ is visualized as a heatmap (main paper, Fig. 9).

**Implementation notes.** We use the benchmark table and compute (i) $S_{m,c}$, (ii) $G_m$, (iii) residuals by ordinary least squares, and (iv) Pearson correlations on residuals. We also report Spearman $\rho$ in auxiliary analysis to confirm robustness to monotonic transformations.

### C.3.2. DOMAIN-WISE SCORE DISTRIBUTIONS (BOXPLOTS)

To complement mean performance summaries, we report domain-wise score distributions across models using boxplots: **task-level** distributions, where each point corresponds to a task mean score within a domain (capturing cross-task variability).

*Figure 8.* Domain-wise score distributions across 9 models (red dashed line separates baselines and VBVR-Wan2.2).

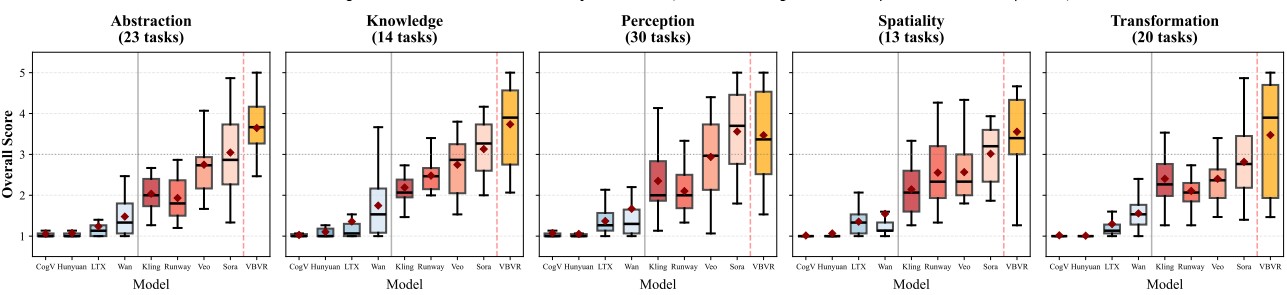

### C.4. Full VBench-I2V Results

*Table 8.* Comprehensive evaluation results on VBench-I2V.

| Model | Total Score | I2V Score | Quality Score | Video-Text Camera Motion | Video-Image Subject Consist. | Video-Image Backgr. Consist. | Subject Consistency | Background Consistency | Motion Smoothness | Dynamic Degree | Aesthetic Quality | Imaging Quality |
|---|---|---|---|---|---|---|---|---|---|---|---|---|
| Wan2.2-I2V-A14B | 0.8816 | 0.9582 | 0.8050 | 0.5444 | 0.9752 | 0.9903 | 0.9468 | 0.9672 | 0.9832 | 0.5285 | 0.6153 | 0.7036 |
| VBVR-Wan2.2 | 0.8835 | 0.9678 | 0.7992 | 0.6592 | 0.9804 | 0.9921 | 0.9547 | 0.9722 | 0.9852 | 0.4106 | 0.6153 | 0.7080 |

### C.5. Cross-Architecture Scaling and Architectural Bottlenecks

**Cross-architecture validation.** To check that the saturation pattern reported in Section 5.2 is not specific to Wan2.2, we replicate the data-scaling protocol on LTX-2.3 (HaCohen et al., 2026), a multimodal dual-stream transformer that jointly generates audio and video via cross-attention between the two streams and employs hierarchical upscaling for efficient high-resolution generation. As shown in Table 9, both ID and OOD performance improve rapidly in the early scaling regime, but the curve flattens beyond 200K–300K samples with only marginal gains thereafter, mirroring the Wan2.2 behavior in Table 3. This consistency across two architecturally distinct DiT backbones reinforces that the observed plateau reflects **a property of the current video-generation paradigm** rather than an idiosyncrasy of a single model.

**Architectural bottlenecks and inductive biases.** The fact that scaling saturates even on in-domain data suggests structural limitations beyond data quality. One concrete suspect is the text-conditioning pathway: today's video diffusion models typically rely on encoder-only language models such as T5 (Raffel et al., 2020) to embed prompts, which may struggle to capture the fine-grained, compositional cues that govern reasoning constraints (e.g., ordering, counting, conditional rules). With the recent emergence of *unified* understanding-and-generation models—OmniGen2 (Wu et al., 2026), Bagel (Deng et al., 2025)—we believe it is timely to equip video reasoners with stronger semantic front-ends (e.g., VLM-style encoders) so that prompt understanding becomes a first-class signal rather than a bottleneck. This direction is also consistent with concurrent observations on physics-grounded video generation that scaling alone does not close the world-modeling gap (Kang et al., 2025).

### C.6. Capability Correlation: Full Analysis

We compute category-level mean scores per model and regress out a model-level *general factor* (overall mean score) before measuring Pearson correlations on the residuals. Figure 9 reveals non-trivial structure among the cognitive dimensions. We

*Table 9.* Cross-architecture scaling validation on LTX-2.3. We finetune LTX-2.3 with progressively larger subsets of VBVR-Dataset (0K–500K samples) and report In-Domain (ID), Out-of-Domain (OOD), and Overall scores on VBVR-Bench. Performance rises rapidly through 200K–300K and then plateaus, mirroring the saturation pattern observed for Wan2.2 in Table 3, indicating that the observed bottleneck is not specific to a single architecture.

| # Data | ID | OOD | Overall |
|---|---|---|---|
| **0K** (LTX-2.3 base) | 0.390 | 0.319 | 0.355 |
| **100K** | 0.458 | 0.335 | 0.397 |
| **200K** | 0.570 | 0.446 | 0.507 |
| **300K** | **0.592** | 0.430 | 0.511 |
| **400K** | 0.591 | 0.443 | 0.516 |
| **500K** (VBVR-LTX2.3) | 0.580 | **0.453** | **0.516** |

*Figure 9.* Residualized capability correlation among five cognitive dimensions across 9 models (Pearson $\rho$). We regress out a model-level general factor (overall strength) to highlight *structural* dependencies and inter-relations.

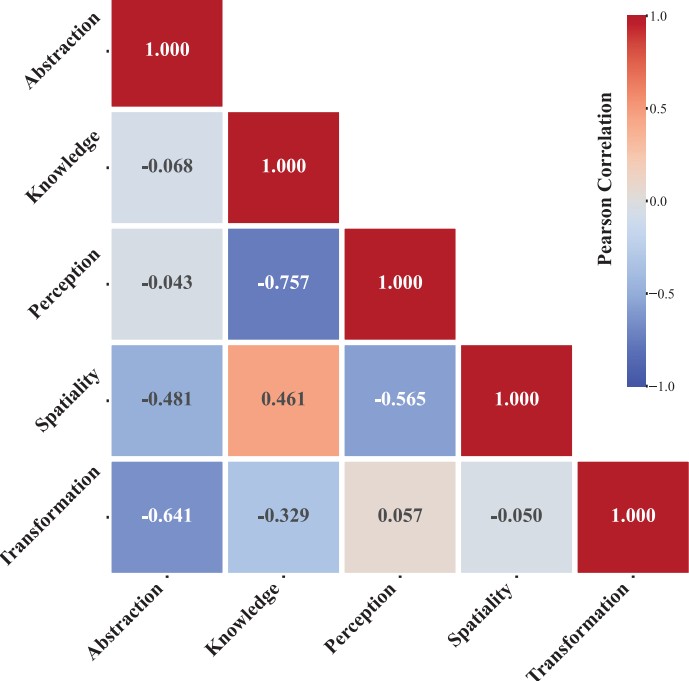

note that the small sample of models ($n=9$) warrants caution in interpreting individual coefficients; we focus on the broader pattern of couplings and trade-offs rather than precise magnitudes.

We observe a positive coupling between **Knowledge** and **Spatiality** ($\rho = 0.461$). This echoes a well-documented connection in cognitive neuroscience: hippocampal place and grid cells, originally characterized as a spatial navigation system (O'Keefe & Dostrovsky, 1971; Hafting et al., 2005), are increasingly understood to support domain-general relational and conceptual learning (Kumaran et al., 2007; 2009; Baram et al., 2024). Tolman's (1948) "cognitive map" hypothesis—that spatial representations are co-opted for non-spatial knowledge organization—provides a suggestive framework for understanding why models that develop stronger spatial representations may also benefit on knowledge-dependent tasks (Whittington et al., 2025; Xiao et al., 2025).

In contrast, **Knowledge** correlates negatively with **Perception** ($\rho = -0.757$), suggesting a trade-off in how models allocate representational capacity. This pattern is reminiscent of an ongoing debate about whether core knowledge systems (e.g., intuitive physics, object permanence) should be understood as perceptual or as genuinely conceptual (Spelke & Kinzler, 2007; Bai et al., 2025; Martin & Barense, 2023).

**Abstraction** shows negative correlations with **Transformation** ($\rho = -0.641$) and **Spatiality** ($\rho = -0.481$), and no positive coupling with any other dimension. In humans, abstract reasoning is associated with prefrontal cortex circuits that are

functionally dissociable from the posterior regions supporting spatial and transformational processing (Badre & Nee, 2018; Vaidya & Badre, 2022), offering a possible parallel for this decoupling pattern.

**Perception** trades off with **Spatiality** ($\rho = -0.565$) but is nearly uncorrelated with both **Abstraction** ($\rho = -0.043$) and **Transformation** ($\rho = 0.057$). **Transformation** is likewise uncorrelated with **Spatiality** ($\rho = -0.050$). These patterns do not conform to a simple hierarchy in which "lower-level" capabilities uniformly support "higher-level" ones; instead, they suggest a more complex dependency structure with trade-offs, paralleling the modular yet interactive organization of biological cognitive systems (Anderson, 2007). Overall, VBVR-Bench not only ranks models but enables interpretable diagnosis of how capabilities co-develop or decouple across systems.

### C.7. Rule-Based Evaluator Pseudocode: G-45 Key Door Matching

To illustrate how a VBVR-Bench scorer is specified, we provide pseudocode for the G-45 Key Door Matching evaluator described in Section 4.1. The evaluator consumes the generated video frames, tracks the colored-dot agent, detects keys and doors in the first and last frames, and aggregates four weighted sub-scores. The actual implementation is released as part of the VBVR-Bench toolkit at https://github.com/Video-Reason/VBVR-EvalKit.

---

**Algorithm 1** G-45 Key Door Matching: rule-based scoring (weights: $w_{\text{move}}=0.30$, $w_{\text{key}}=0.35$, $w_{\text{door}}=0.25$, $w_{\text{seq}}=0.10$).

---

1: **Input:** Video frames $V = [f_0, f_1, \ldots, f_{T-1}]$
2: **Output:** Scalar score $s \in [0, 1]$
3: **if** $T < 2$ **then**
4:     **return** 0
5: **end if**
6: $P \leftarrow \text{TRACKAGENT}(V)$ {ordered list of agent centroids}
7: **if** $|P| < 2$ **then**
8:     **return** 0
9: **end if**
10: $K_0 \leftarrow \text{DETECTKEYS}(f_0)$; $K_T \leftarrow \text{DETECTKEYS}(f_{T-1})$; $D_0 \leftarrow \text{DETECTDOORS}(f_0)$
11: *(1) Agent movement (30%)*
12: $d_{\max} \leftarrow \max_{p \in P} \|p - P[0]\|$
13: **if** $d_{\max} < 30\,\text{px}$ **then**
14:     **return** 0 {agent never left start $\Rightarrow$ task fails}
15: **end if**
16: $s_{\text{move}} \leftarrow \text{MOVEMENTBUCKET}(d_{\max})$ {1.0 if $>150$, 0.8 if $>100$, 0.6 if $>60$, 0.3 otherwise}
17: *(2) Key collection (35%): require $\min_p \|p - k\| < 50$ AND key absent in $K_T$*
18: (key_ok, $c_{\text{key}}$, $s_{\text{key}}$) $\leftarrow \text{CHECKKEYCOLLECTED}(K_0, K_T, P, V)$
19: *(3) Door reached (25%)*
20: (door_ok, $c_{\text{door}}$, $s_{\text{door}}$) $\leftarrow \text{CHECKDOORREACHED}(D_0, P, V, \text{key\_ok})$
21: **if** door_ok $\wedge$ key_ok **then**
22:     **if** $c_{\text{key}} = c_{\text{door}}$ **then**
23:         *keep* $s_{\text{door}}$ {matching color}
24:     **else if** $c_{\text{key}} = \bot \vee c_{\text{door}} = \bot$ **then**
25:         $s_{\text{door}} \leftarrow 0.5 \cdot s_{\text{door}}$ {color undetectable; partial credit}
26:     **else**
27:         $s_{\text{door}} \leftarrow 0$ {wrong color}
28:     **end if**
29: **else**
30:     $s_{\text{door}} \leftarrow 0$ {door not reached or no key}
31: **end if**
32: *(4) Sequence (10%): key before door*
33: $s_{\text{seq}} \leftarrow \text{SEQUENCECHECK}(V, K_0, D_0, P)$ if key_ok else 0
34: **return** $w_{\text{move}}s_{\text{move}} + w_{\text{key}}s_{\text{key}} + w_{\text{door}}s_{\text{door}} + w_{\text{seq}}s_{\text{seq}}$

---

Notable design choices follow directly from the task semantics: the hard cutoff at $d_{\max} < 30\,\text{px}$ enforces that scenes

which never move the agent receive 0 (preventing degenerate "snapshot" generations from accumulating partial credit); the color-match check between collected key and reached door realizes the relational constraint that gives the task its name; and the strict ordering test in SEQUENCECHECK ensures that procedural correctness is rewarded only when the key is genuinely collected before the door is approached. Each of the 100 evaluation tasks is paired with an analogous rule, decomposed into 3–4 weighted sub-criteria; the full set is implemented in the released VBVR-Bench toolkit.

# D. Selected Tasks and Rubrics

This appendix includes a curated set of tasks that require multi-step planning and/or multiple interacting constraints. For each task, we show the initial and final frames, followed by the evaluation rubric used to score model outputs.

## D.1. Representative Kept vs. Discarded Tasks

Our task pool starts from over 300 candidate ideas (Section B.1), of which 200 are retained after the dual-review process. Decisions are guided by whether a task admits (i) a deterministic ground truth, (ii) unambiguous visual rendering in short clips, and (iii) a rule-based scorer with bounded variance. We list a few representative decisions below to make the criteria concrete; the full kept/discarded list is released alongside the VBVR-Dataset task taxonomy.

**Kept tasks (examples).**

- **O-19 Mirror Reflection.** Light-beam trajectories follow geometric laws, yielding deterministic, verifiable outcomes that can be checked frame-by-frame against the reflected ray.
- **O-53 Clock Reading and Time Reasoning.** Future positions of clock hands are mathematically unambiguous and easily verified by extracting hand angles in the final frame.

**Discarded tasks (examples).**

- **Elastic Collisions with Friction.** Continuous block trajectories are highly sensitive to initial conditions and integrator choice, making any single rendered trajectory a poor target for deterministic scoring.
- **Human Pose Prediction.** Human motion admits infinitely many valid continuations; the lack of a unique ground truth precludes a verifiable rule.
- **Logic Gate Circuits.** Multi-step signal flows are visually cluttered in short clips and require symbolic tracing that current rule-based scorers cannot disambiguate reliably from rendered video alone.

The complete kept-vs.-discarded list across all 300+ original candidates, together with per-task rationale and reviewer notes, is released alongside the VBVR-Dataset task taxonomy at `https://github.com/Video-Reason/VBVR-EvalKit`.

**Stable Sort (Task G-3)**   Rearrange objects by grouping them by shape type and sorting each group by size while preserving all attributes, testing multi-constraint spatial reasoning, attribute fidelity, and rule following.

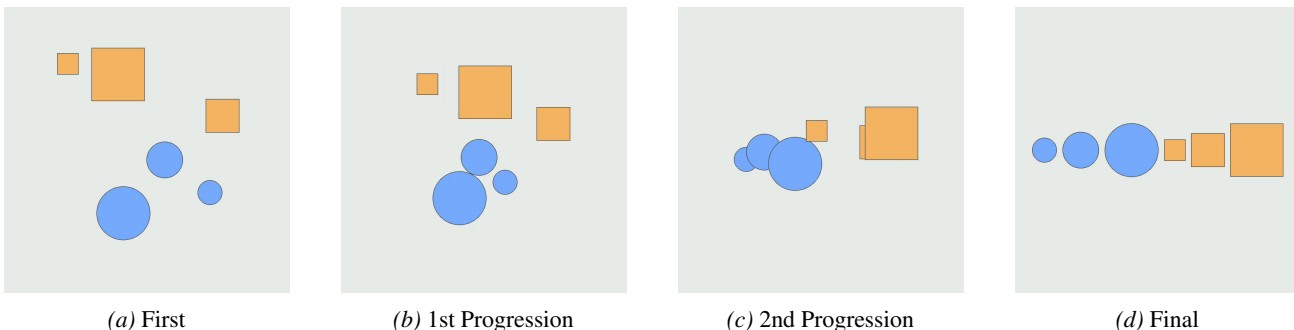

*(a)* First   *(b)* 1st Progression   *(c)* 2nd Progression   *(d)* Final

*Figure 10.* Stable Sort

**Example Prompt** "The scene contains two types of shapes, each type has three shapes of different sizes arranged randomly. Keep all shapes unchanged in appearance (type, size, and color). Only rearrange their positions: first group the shapes by type, then within each group, sort the shapes from smallest to largest (left to right), and arrange all shapes in a single horizontal line from left to right."

*Human Annotation Scoring (1–5):*

- **5 (Perfect).** Correct grouping; correct within-group ascending size order (left to right); horizontal alignment; attributes preserved (type/size/color); reasonable spacing, no overlaps.

- **4 (Near-perfect).** Correct grouping and ordering with one minor imperfection (e.g., small color deviation (hue shift within ±10%), small size deviation (within ±5%), slight vertical misalignment, or mildly uneven spacing) while the intended layout remains clear.

- **3 (Partially correct).** Grouping is correct but ordering is wrong in at least one group; *or* grouping and ordering are correct but attribute changes are noticeable (e.g., color shift > 10%, size change > 5%, or mild deformation while still recognizable).

- **2 (Multiple errors).** Incorrect grouping and/or incorrect ordering, and/or missing/extra objects.

- **1 (Failure).** Objective not achieved (no intended grouping/ordering) and/or severe object modification/loss.

*Evaluation dimensions (automated scorer weights):*

- **Classification (30%).** Require at least 6 detected shapes and at least 2 color groups. Sort the shapes by horizontal position and count color transitions; the score is 1.0 if transitions $\leq$ #groups $-$ 1, else decays by 0.3 per excess transition. If fewer than 6 shapes are detected, scale down by #detected/6 × 0.5.

- **Ordering (30%).** Within each color group, sort shapes by $x$-position and compute the fraction of consecutive pairs $(s_i, s_{i+1})$ satisfying $\text{area}(s_i) < \text{area}(s_{i+1})$; average across groups.

- **Object fidelity (30%).** $0.4 \cdot \text{count\_match} + 0.3 \cdot \text{total\_area\_ratio} + 0.3 \cdot \text{type\_list\_match}$, where $\text{count\_match} = 1 - |n_{\text{init}} - n_{\text{final}}| / \max(n_{\text{init}}, 1)$, $\text{area\_ratio} = \min(A_{\text{init}}, A_{\text{final}}) / \max(A_{\text{init}}, A_{\text{final}})$, and type\_list\_match is the fraction of matched entries between sorted initial/final shape-type lists.

- **Layout (10%).** $\max(0, \ 1 - \text{Var}(y\text{-coordinates})/5000)$, rewarding horizontally aligned final shapes.

**Grid Avoid Obstacles (Task G-15)** Navigate a $10 \times 10$ grid from start to goal using only 4-neighbor moves without entering obstacle cells, testing shortest-path planning under hard constraints.

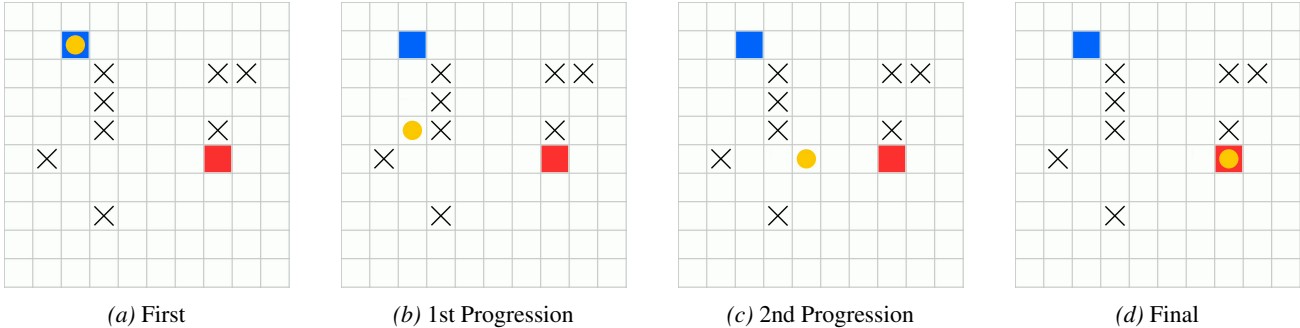

| *(a)* First | *(b)* 1st Progression | *(c)* 2nd Progression | *(d)* Final |

*Figure 11.* Grid Avoid Obstacles

**Example Prompt**: "The scene shows a 10x10 grid with a blue start square (containing a yellow circular agent), a red end square, and multiple black X marks indicating obstacles. Starting from the blue start square, the agent can move to adjacent cells (up, down, left, right) each step. The goal is to move the agent to the red end square along the shortest path without entering any cells marked with black X obstacles."

*Human Annotation Scoring (1–5):*

- **5 (Perfect).** Reaches the goal; avoids all obstacles; uses a shortest (or tied-shortest) obstacle-avoiding path; movement is legal (4-neighbor, within-grid) and agent appearance is preserved.

- **4 (Near-perfect).** Reaches the goal and avoids all obstacles; only a minor imperfection (e.g., $\leq 2$ extra steps vs. optimal, slight appearance drift, or minor legal-motion jitter).

- **3 (Partially correct).** Clear attempt but with a notable issue (e.g., enters exactly one obstacle cell once, $> 2$ extra steps, stops short of the goal, or an occasional illegal/diagonal tendency).

- **2 (Mostly incorrect).** Multiple violations (e.g., enters 2–3 obstacle cells) and/or out-of-grid motion and/or highly inefficient/random routing and/or fails to reach the goal.

- **1 (Failure).** No meaningful progress or unrelated output; frequent obstacle crossings ($\geq 4$) or the grid is corrupted.

*Evaluation dimensions (automated scorer weights):*

- **Task completion (45%).** Agent reaches the red end cell. Full credit if the agent's final position is within 40 px of the endpoint; partial credit (0.3) if within 80 px; otherwise 0.

- **Grid preservation (30%).** The blue start cell and red end cell must retain their colors throughout. *Any* unauthorized color change to these cells causes the task to fail outright (score = 0).

- **Obstacle avoidance (15%).** $1 - \text{collision\_count}/\text{trajectory\_length}$, where a collision is the agent coming within 30 px of any black-X obstacle.

- **Step-by-step movement (10%).** Penalizes frames where the agent jumps more than 100 px between consecutive positions.

**Grid Go Through Block (Task G-16)**    Plan a near-shortest 4-neighbor route that visits all marked target cells (in blue) before reaching the goal (in red) in a $10 \times 10$ grid, testing multi-goal route optimization under movement constraints.

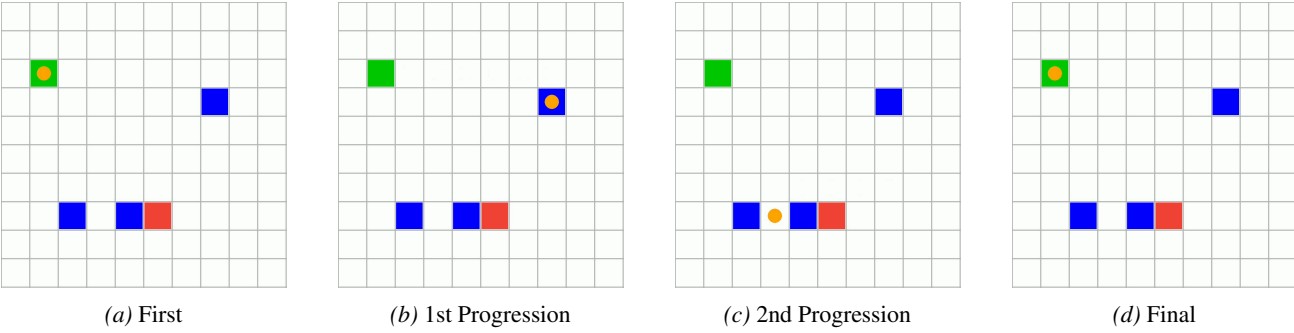

| *(a)* First | *(b)* 1st Progression | *(c)* 2nd Progression | *(d)* Final |

*Figure 12.* Grid Go Through Block

**Example Prompt** :"The scene shows a 10x10 grid with a green start square (containing an orange circular agent), a red end square, and multiple blue rectangular blocks. Starting from the green start square, the agent can move to adjacent cells (up, down, left, right) each step. The goal is to move the agent to the red end square along the shortest path that passes through all blue blocks (the agent must visit every blue block before reaching the red end square)."

*Human Annotation Scoring (1–5):*

- **5 (Perfect).** Starts at the start cell and ends at the goal; visits all targets; uses a globally shortest route under the visit-all constraint; motion is legal (4-neighbor, within-grid) and agent appearance is preserved.

- **4 (Near-perfect).** Reaches the goal and visits all targets; only small imperfections (e.g., $\leq 3$ extra steps, minor appearance drift, or slight presentation issues while the realized path remains grid-orthogonal).

- **3 (Partially correct).** Clear attempt but with a notable issue (e.g., misses exactly one target, $> 3$ extra steps, stops before the goal, or an occasional illegal move).

- **2 (Mostly incorrect).** Misses two or more targets and/or fails to reach the goal, and/or frequent illegal/out-of-grid motion, and/or highly inefficient/random routing.

- **1 (Failure).** No meaningful progress; ignores targets or output is unrelated / grid is broken.

*Evaluation dimensions (automated scorer weights):*

- **Block visit (40%).** Fraction of blue blocks the agent's trajectory passes within 40 px of: $|\text{visited\_blocks}|/|\text{all\_blocks}|$.

- **Path optimality (30%).** Min/max ratio of generated path length vs. ground-truth path length, where lengths are summed Euclidean distances between consecutive detected agent positions.

- **Task completion (20%).** Agent reaches the red endpoint in the final frame. Full credit if within 50 px; otherwise $\max(0, 1 - \text{dist}/100)$. (Note: only the final position is checked; the scorer does not verify the full start→targets→end ordering.)

- **Movement (10%).** Penalizes diagonal moves (consecutive positions where $|\Delta x| > 10$ *and* $|\Delta y| > 10$): $\max(0, 1 - 0.2 \cdot \text{diagonal\_count})$.

**Directed Graph Navigation (Task G-31)**    Navigate from the start node to the goal node by traversing only along directed edges (respecting arrow direction) with a minimum-hop path, testing goal-directed graph planning under directionality constraints.

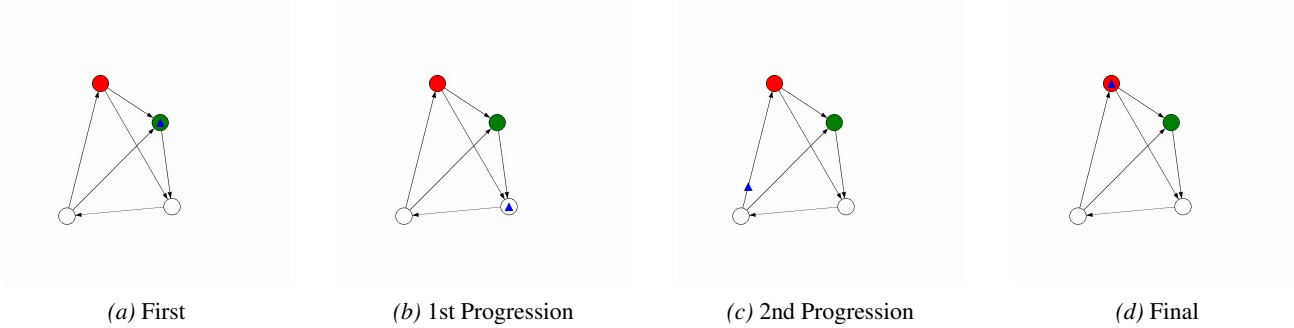

| (a) First | (b) 1st Progression | (c) 2nd Progression | (d) Final |

*Figure 13.* Directed Graph Navigation

**Prompt** :"The scene shows a network of nodes connected by directed edges (edges with arrows indicating direction) with a green starting node, a red ending node, and a blue triangular agent positioned at the green starting node. The agent can only move along edges in the direction they point (from the source node to the target node, cannot move backwards), moving from one node to an adjacent node each step. Move the blue triangular agent from the green starting node to the red ending node along the path with the minimum number of steps."

*Human Annotation Scoring (1–5):*

- **5 (Perfect).** Reaches the goal; uses a minimum-hop path; strictly follows arrow directions and moves only along existing edges; motion is continuous and the graph is preserved.

- **4 (Near-perfect).** Reaches the goal with correct direction/edge-following behavior; only minor presentation issues while path length remains shortest.

- **3 (Partially correct).** Reaches the goal but with a notable issue (e.g., +1–2 extra hops, a single direction violation, or slight deviation from drawn edges).

- **2 (Mostly incorrect).** Severe issues: $\geq 3$ extra hops, multiple direction violations, "flying"/cutting across space, reaches the wrong node, or alters graph structure.

- **1 (Failure).** Does not reach the goal, behavior is random/invalid (ignores arrows), or the graph is corrupted.

*Evaluation dimensions (automated scorer weights):*

- **Circle color preservation (50%).** The green start node and red end node must retain their colors. If the combined green+red pixel count changes by more than 100% relative to the initial frame, the task fails outright (score $= 0$). Otherwise scored as $\max(0, 1 - \text{total\_change})$.

- **Task completion (35%).** Agent (blue triangle) reaches the red end node. Full credit if final agent position is within 50 px of the endpoint; partial credit (0.3) if within 100 px; otherwise 0.

- **Path quality (15%).** Penalizes large jumps ($> 150$ px) between consecutive frames, reflecting smooth motion rather than teleportation.

**Key Door Matching (Task G-45)**    In a maze, collect a key and reach the same-colored door via legal corridor moves near shortest, testing color-relational binding, sequencing, and constrained navigation.

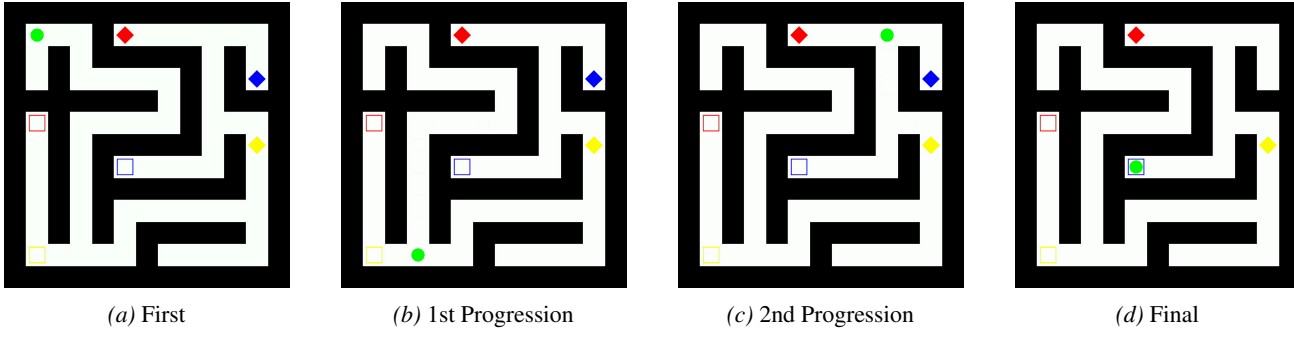

*(a)* First             *(b)* 1st Progression             *(c)* 2nd Progression             *(d)* Final

*Figure 14.* Key Door Matching

**Prompt** :"The scene shows a maze with a green circular agent, colored diamond-shaped keys, and colored hollow rectangular doors. Find the Blue key and then navigate to the matching Blue door, showing the complete movement process step by step."

*Human Annotation Scoring (1–5):*

- **5 (Perfect).** Reaches a key, then reaches a same-colored door (correct order); path is legal and continuous; path length is near-optimal (e.g., $\leq 120\%$ of BFS shortest for start$\to$key$\to$door); key disappears when reached; full trajectory is shown.

- **4 (Near-perfect).** Reaches a key and a same-colored door in correct order; minor issues only (e.g., slightly longer path $\sim 120\%$–$150\%$, small jitter/backtracking, or a 1–2 frame delay in key disappearance).

- **3 (Partially correct).** Reaches a key but makes a major mistake (e.g., goes to a non-matching door, completes only one stage, noticeably inefficient path $\sim 150\%$–$200\%$, or 1–2 minor wall violations).

- **2 (Mostly incorrect).** Mismatched key/door colors and/or wrong order (door before key), and/or multiple wall crossings, and/or extremely inefficient path ($> 200\%$), or stops at irrelevant locations.

- **1 (Failure).** Agent barely moves or moves randomly; ignores maze constraints; mismatched key and door colors; or agent behavior is abnormal.

*Evaluation dimensions (automated scorer weights, consistent with Section 4.1 and Algorithm 1):*

- **Agent movement (30%).** Agent must physically move away from the starting position. Score based on the maximum distance reached from start: 0 if $< 30\,\text{px}$ (the agent never left start, complete failure); 0.3 if $< 60\,\text{px}$; 0.6 if $< 100\,\text{px}$; 0.8 if $< 150\,\text{px}$; 1.0 otherwise.

- **Key collection (35%).** Agent must physically reach a key's position (within 50 px) *and* that key must subsequently disappear. Score 1.0 if both conditions hold; 0.6 if the agent reached the key but the key remained; 0 if a key disappeared without the agent ever reaching it (anti-cheat).

- **Door reached (25%).** After collecting a key, the agent must reach a door's position. The collected key color *must match* the door color, otherwise the score is 0. Distance buckets give 1.0 ($< 40\,\text{px}$), 0.7 ($< 70\,\text{px}$), 0.4 ($< 100\,\text{px}$).

- **Sequence (10%).** The earliest key visit must precede the earliest door visit. Score 1.0 if the key was visited first; 0.2 if the door was visited first; 0 if no key was ever collected.

