# OpenReview forum: "A Very Big Video Reasoning Suite"
_ICML.cc/2026/Conference — ICML 2026 regular_

### Official Review · Reviewer_JC8N · 2026-03-09

**Soundness:** 4
**Presentation:** 3
**Significance:** 4
**Originality:** 4
**Overall Recommendation:** 5
**Confidence:** 3

**Summary:**

The paper introduces VBVR, a massive dataset with 200 curated reasoning tasks and over one million video clips approximately three orders of magnitude larger than existing datasets for studying video (generation) reasoning. Along with that the authors also propose VBVR-Bench, a benchmark and evaluation framework for video generation which is rule-based, and statistically verified with human evaluators. They also provide benchmarking on various open-source and proprietary video generation models, a baseline finetuned on the proposed dataset and a scaling analysis of its training. The authors claim that this work can be a foundation for future research in this domain.

**Compliance With Llm Reviewing Policy:**

Affirmed.

**Final Justification:**

My concerns were mostly addressed. I am maintaining my score.

**Key Questions For Authors:**

1. Does any of the tasks involve “real world” style or videos?
2. Are the rules different for all tasks? Can the authors provide an analysis of how many rules repeat how frequently/overlap for different tasks?

**Limitations:**

yes

**Strengths And Weaknesses:**

**Strengths:**
1. **Massive dataset**: The size of the proposed dataset is very large, especially compared to existing datasets in the domain which is a valuable contribution to the community.
2. **Statistically verified evaluation protocols**: The rule-based evaluation protocols proposed by the authors is in accordance with human evaluators which they provide statistical evidence for
3. **Extensive benchmarking and analysis**: The authors show extensive benchmarking on open-source and proprietary video generation models.
4.  **Scaling Analysis**: The authors provide a detailed scaling analysis of proposed baseline training
5. **Solid Baseline**: The baseline proposed by training Wan2.2 is solid as it is outperforming the best proprietary models

**Weaknesses:**
1. **More details about original Task Pool required**: The appendix provides some information about initial Task Pool, but it is still lacking details as to how the tasks were conceptualized. An end-to-end example of a particular category with a list of all the candidate questions and then kept and discarded questions out of them with an analysis (within limit of review space) might be helpful.
2. **Slightly misleading title**: Video Reasoning generally refers to the capacity of models to interpret reasoning in existing videos. Here the suite is testing a model's capacity of video generation with a given understanding.
3. **Initial performance concern**: The trained baseline (and top proprietary models) are already performing so well. Even if they are long way off from Human performance, but high performance of baseline can be concerning as models in the near future can potentially close the gap quickly.
4. **Complicated writing style**: Due to the dense nature of the paper the writing can get complicated at some points which makes the paper difficult to understand.

**Minor Weaknesses:**
1. Authors need to clarify this line “most existing video reasoning benchmarks provide few or no video samples”.
2. Space could have been managed better in formatting the paper, as there are empty spaces between paragraphs

---

> ### Author Rebuttal · Authors · 2026-03-31
>
> We want to sincerely thank you for insightful positive evaluation, as well as for providing constructive feedback that help us further clarify and improve our work.
>
> **Weakness 1: More details on the original task pool**
>
> Due to word limit, we present two representative tasks that are retained and three that are discarded; the full list will be included in the revision.
>
> Kept Tasks
> - O-19 Mirror Reflection: Light beam trajectories follow geometric laws, yielding deterministic, verifiable outcomes.
> - O-53 Clock Reading and Time Reasoning: Future positions of clock hands are mathematically unambiguous and easily verified.
>
> Discarded Tasks
> - Elastic Collisions with Friction: Continuous block trajectories are too sensitive for deterministic scoring.
> - Human Pose Prediction: Human motion admits infinitely many valid outcomes, lacking a unique ground truth.
> - Logic Gate Circuits: Multi-step signal flows are visually cluttered in short clips, preventing clear evaluation.
>
>
> **Weakness 2: “Video reasoning” terminology**
>
> While “video reasoning” often denotes reasoning *over* input videos, we adopt a broader (and we argue more fundamental) definition: reasoning *through* video, where video is the substrate of intelligence. Specifically, we focus not only on video understanding, but also spatiotemporally consistent output videos that satisfy physical and logical rules. This perspective aligns with recent works that view generation as a vehicle for reasoning (e.g., [1,2]). To avoid ambiguity, we will clarify this in the revision.
>
> **Weakness 3: Baseline performance and remaining gap:**
>
> The gap to human-level reasoning not only still remains substantial (~30 points), but closing the remaining gap can be significantly more difficult. In particular, we observe consistent failure modes in current models that are not solvable by scaling alone.
>
> Moreover, we view benchmark saturation as a natural and expected outcome. Historically, impactful benchmarks (e.g., ImageNet, COCO) have eventually been saturated, yet they continue to serve as catalysts for progress. In this sense, VBVR-Bench is designed as a foundation that isolates core spatiotemporal reasoning under controlled settings, upon which more challenging and realistic benchmarks can be built.
>
> **Weakness 4 and Minor Weakness 2:**
>
> Thank you for the feedback on writing. We will revise to improve clarity, presentation, and formatting as you suggested.
>
> **Minor Weakness 1:**
>
> As summarized in Table 1, prior “video reasoning” benchmarks (e.g., Video-Zero-Shot, V-ReasonBench), while designed to probe temporal or causal reasoning, provide little to no actual video data. As a result, they lack ground-truth videos for fine-grained, per-frame evaluation, and the absence of training video data creates a mismatch between the intended tasks and the available supervision. In contrast, VBVR provides large-scale video data for both training and evaluation across diverse reasoning tasks, enabling models to learn and be evaluated directly in the video domain.
>
> **Question 1: Real-world transfer**
>
> VBVR intentionally uses synthetic data to isolate reasoning from visual quality (e.g., texture, lighting, aesthetics), enabling controlled and precise evaluation.
>
> Interestingly, VBVR-Wan2.2 demonstrates transfer to real-world scenarios. On VBench, a real-world video benchmark, model improves controllability and consistency (e.g., Video-Text Camera Motion: $0.54 \to 0.65$), along with gains in Subject and Background Consistency. Trade-off in unconstrained motion (Dynamic Degree) is expected due to strict adherence to logical prompts.
>
> |Model|Total Score|Video-Text Camera Motion|Video-Image Subject Consistency|Video-Image Background Consistency|Dynamic Degree|
> |:-|:-:|:-:|:-:|:-:|:-:|
> |Wan2.2-I2V|0.8816|0.5444|0.9752|0.9903|0.5285|
> |VBVR-Wan2.2|0.8835|0.6592|0.9804|0.9921|0.4106|
>
> Qualitative examples (https://github.com/engineerAnonymous/2026_VBVR_rebuttal) also show that VBVR avoids invalid behaviors (e.g., walking through walls) and better follows real-world instructions while improving background consistency and reducing visual artifacts.
>
> Finally, recent work [3] further validates the reasoning capabilities on both synthetic and real-world cases, providing additional evidence of sim-to-real transfer.
>
> **Question 2: Evaluation rules**
>
> Rules are task-specific, with 100 rules for 100 tasks defining distinct constraints and success criteria. For illustration, a detailed example of *Task G-45 Key Door Matching* is provided in Sec. 4.1, along with abstract semantic descriptions of rules in Sec. D. Due to word limit, pseudocode examples will be provided in the revised appendix.
>
>
> [1] Chen et al., TiViBench: Benchmarking Think-in-Video Reasoning for Video Generative Models, arXiv 2025.
>
> [2] Yang et al., Reasoning via Video: The First Evaluation of Video Models' Reasoning Abilities through Maze-Solving Tasks, arXiv 2025.
>
> [3] Wang et al., Demystifying Video Reasoning, arXiv 2026.

---

> > ### Author Rebuttal · Reviewer_JC8N · 2026-04-03
> >
> > The authors have addressed most of my concerns.
> >
> > For weakness 2, I appreciate the authors' response, but I do not fully agree with the justification provided for calling it "video reasoning".
> >
> > I would also like another example of "Real-world transfer" relating to Question 1.
> >
> > Rest everything is fine for me. I will most likely be maintaining my positive score.

---

> > > ### Author Response · Authors · 2026-04-05
> > >
> > > Thank you for your thoughtful feedback and for your positive assessment of our revisions.
> > >
> > > We are glad that our responses have addressed most of your concerns. Regarding Weakness 2, we appreciate your perspective on the use of the term “video reasoning.” We will further think about it on how to avoid potential ambiguity.
> > >
> > > For your request on “real-world transfer” (Question 1), we have added additional examples (Examples 5–8) in the anonymous repository(https://github.com/engineerAnonymous/2026_VBVR_rebuttal) to better illustrate.
> > >
> > > Thank you again for your constructive feedback. We appreciate your consideration and are encouraged by your inclination to maintain a positive score.

---

### Official Review · Reviewer_zjGY · 2026-03-10

**Soundness:** 4
**Presentation:** 3
**Significance:** 4
**Originality:** 4
**Overall Recommendation:** 4
**Confidence:** 4

**Summary:**

This paper tackles the critical bottleneck of data scarcity and evaluation ambiguity in video-based reasoning for generative models. While recent advancements have significantly improved the visual quality of video generation, systematic evaluation of spatiotemporal reasoning remains constrained by the lack of large-scale, verifiable datasets. To bridge this gap, the authors introduce the Very Big Video Reasoning (VBVR) suite, comprising a massive programmatic dataset and a deterministic evaluation benchmark.

**Compliance With Llm Reviewing Policy:**

Affirmed.

**Final Justification:**

Thanks for the clear rebuttal of the authors, my concerns have been addressed very well. I believe its a solid work.

**Key Questions For Authors:**

1. Concrete Details on Quality Control: The paper mentions a fully automated quality control process during large-scale data generation. Could the authors elaborate more concretely on the implementation? Specifically, how are visual compliance, boundary constraints, and edge cases deterministically verified without human intervention?

2. OOD Generalization to New Visual Domains: The current OOD evaluation primarily focuses on unseen reasoning tasks. Have the authors considered extending this to entirely new visual domains (e.g., complex 3D embodied AI scenes) to better stress-test the models' visual robustness?

3. Sim-to-Real Transferability: Following up on the synthetic nature of the dataset, is there any evidence showing that models fine-tuned on VBVR-Dataset can generalize their reasoning capabilities to real-world video benchmarks? Discussions or preliminary results on this sim-to-real transfer would significantly strengthen the paper.

**Limitations:**

Please see the weakness and questions.

**Strengths And Weaknesses:**

Strengths:

1. Timely & Crucial Problem: This work addresses the critical but underexplored bottleneck of video reasoning. It provides the essential data and benchmarks needed to advance this field, paralleling the progress seen in text-based reasoning.

2. Sound & Scalable Data Curation: The dataset construction is highly sound, featuring a well-designed task taxonomy. The proposed automatic generation mechanism makes high-quality data creation deterministically scalable.

3. Valuable Empirical Insights: Based on the proposed benchmark, the paper conducts extensive empirical studies. The findings on model scaling behaviors and the structural correlations between different reasoning tasks offer significant value to the community.

Weaknesses
Synthetic Data Limitation: The dataset relies entirely on synthesized, rendered videos (e.g., puzzles and grid games). It lacks the unstructured complexity of real-world videos, which are ultimately necessary to evaluate authentic spatial-temporal reasoning and broad world knowledge.

---

> ### Author Rebuttal · Authors · 2026-03-31
>
> We want to thank you for the positive assessment and insightful questions regarding data quality, synthetic design, and real-world generalization.
>
> **Question 1: Concrete details on automated quality control**
>
> In production we run fully automated quality-control at scale (millions of samples), as summarized in Sec. 3.3.3: every sample passes a shared validation process before it is kept, which verifies that a valid solution exists, that rendered frames satisfy task-specific visual and layout rules, and that all sampled parameters stay within each generator’s declared ranges; failed cases are retried automatically, and repeated failures are logged for manual inspection and follow-up. We unpack visual compliance, boundary constraints, and edge cases below as you suggested, and the details will be included in the revised version.
>
> 1) **Visual compliance** is enforced with deterministic, generator-defined rules on the rendered frames, because our benchmark scoring is rule-based and needs unambiguous visuals. Concretely, validators reject layouts where objects overlap beyond each task’s allowed tolerance: the initial frame is expected to carry all information needed for that instance, so heavy overlap would occlude shapes (e.g., in ordering or multi-object layouts) and make boundaries indeterminate for models that only see the first image. When tasks render text or symbolic marks on the canvas, we also enforce minimum font or stroke sizes so those on-frame cues stay legible for video models.
>
> 2) **Boundary constraints** are enforced by combining a parameter-bounds pass (each field must stay inside the ranges configured for that task) with validity constraints produced during sampling, For example, for navigation-style tasks we require solvable worlds and verify solvability with the same search-style checks we use in development (e.g., pathfinding). Representative generator configs use bounded discrete ranges such as grid sizes from 5 to 10 (inclusive) and object counts from 3 to 8 for maze-style instantiations, which keeps the sampled space aligned with what the renderer and evaluator expect.
>
> 3) **Edge cases** are handled in two layers. Task admission already avoids structural pathologies we know break rendering or evaluation (for example boundary overflow off the canvas or goals that are initially fully occluded). When rare runtime faults still occur, the production workflow described in Sec. 3.3.3 retries, logs persistent failures, and tracks aggregate validation failure rates so we can patch generators, again without manual review of every clip.
>
> **Weakness, Question 2 & 3: Synthetic data, visual robustness and sim-to-real transfer**
>
> We agree that extending to more complex visual domains is important. VBVR is intentionally designed with synthetic data to decouple visual quality (e.g., texture, lighting, aesthetics) from reasoning ability, enabling precise evaluation of spatiotemporal reasoning under controlled conditions. We believe VBVR can inspire the development of more challenging and realistic video reasoning benchmarks, such as those in 3D embodied AI scenarios, which require integrated capabilities in perception, object recognition, and action execution.
>
> Below, we provide evidence that the model trained on VBVR demonstrates visual robustness and knowledge transferability in various real-world scenarios.
>
> First, we evaluate VBVR-Wan2.2 on VBench, a real-world video benchmark, and observe clear improvements in controllability and consistency. In particular, Video-Text Camera Motion improves from $0.54 \to 0.65$, along with gains in Subject and Background Consistency. While Dynamic Degree decreases, we view this as an expected trade-off, as VBVR emphasizes strict adherence to logical prompts and precise local edits, which may limit unconstrained motion.
>
> |Model|Total Score|Video-Text Camera Motion|Video-Image Subject Consistency|Video-Image Background Consistency|Dynamic Degree|
> |:-|:-:|:-:|:-:|:-:|:-:|
> |Wan2.2-I2V|0.8816|0.5444|0.9752|0.9903|0.5285|
> |VBVR-Wan2.2|0.8835|0.6592|0.9804|0.9921|0.4106|
>
> Second, we gather qualitative examples of real-world scenarios at https://github.com/engineerAnonymous/2026_VBVR_rebuttal in maze-like scenarios, the model correctly avoids invalid behaviors such as “walking through walls,” and in real-world prompts (e.g., ask a person to constantly walk forwards camera), it better follows instructions while maintaining improved background consistency and fewer visual artifacts. These results suggest that VBVR primarily enhances structural reasoning and controllability, which can transfer to more realistic settings despite the absence of photorealistic training data.
>
> Finally, recent work [1] further validates the reasoning capabilities of VBVR-Wan2.2 on both synthetic and real-world cases, providing additional evidence of sim-to-real transfer.
>
> [1] Wang et al., Demystifying Video Reasoning, arXiv 2026.

---

> > ### Author Rebuttal · Reviewer_zjGY · 2026-04-04
> >
> > Thanks for the clear rebuttal of the authors, my concerns have been addressed very well. I believe its a solid work.

---

> > > ### Author Response · Authors · 2026-04-04
> > >
> > > We sincerely thank you for the insightful comments. We greatly appreciate your positive assessment and the recognition of the strength of our work.

---

### Official Review · Reviewer_v9vY · 2026-03-11

**Soundness:** 3
**Presentation:** 3
**Significance:** 3
**Originality:** 3
**Overall Recommendation:** 3
**Confidence:** 4

**Summary:**

This paper introduces the VBVR (Very Big Video Reasoning) suite, which features the VBVR-Dataset consisting of 200 meticulously designed reasoning tasks and over one million video clips. The authors also propose VBVR-Bench. By conducting large-scale training and analysis on models such as Wan-2.2 using the VBVR-Bench, the authors reveal the Scaling Effects of video reasoning capabilities, as well as generalization characteristics across in-domain and out-of-domain tasks.

**Compliance With Llm Reviewing Policy:**

Affirmed.

**Final Justification:**

We thank the authors for the detailed rebuttal. While the response provides useful clarifications, our main concerns remain only partially addressed.

For Q1, our concern is whether the claimed scaling gap could be mitigated by incorporating richer real-world data. The rebuttal instead argues that the model fails to fully fit the synthetic distribution, suggesting the issue is not data diversity. However, this does not directly address our point.

For Q2, it is still unclear whether the bottleneck arises from limited generative quality or reasoning capability, as no targeted ablations are provided to disentangle these factors.

Therefore, I maintain my Overall Recommendation as Weak reject.

**Key Questions For Authors:**

1. Does the observed performance plateau also occur in other architectures?

2. Does the performance plateau reflect an absolute limit of reasoning, or simply the saturation of the model fitting the relative simple  synthetic data?

3. How can the evaluation metric reflect the deficiency in the generation capability or the reasoning capability?

**Limitations:**

Yes.

**Strengths And Weaknesses:**

[+] The scale and diversity of the data are highly valuable for the future development of video generation and reasoning models.

[+] The authors' commitment to open-sourcing the data, toolkit, and models is crucial for facilitating community reproduction and follow-up research.

[-] The authors claim in the Introduction that there exists a "persistent gap between model and human performance that cannot be bridged by data scaling alone." While the empirical results show a performance plateau, I find this statement to be a potential overclaim. Firstly, the scaling experiments were conducted exclusively on Wan-2.2, meaning the scaling laws for other model architectures have not been demonstrated. Secondly, although the data used for VBVR training reached a scale of 1M, these samples are essentially synthetic data, whose upper bounds of diversity and complexity are dictated by the "generation scripts." The so-called "gap that cannot be bridged by scaling" might actually result from the diminishing marginal utility of synthetic data once it reaches a certain scale. If higher-quality, more logically challenging real-world long video data were incorporated, the scaling law might exhibit a different slope.

[-] The paper primarily relies on rule-based scorers for evaluation, which raises a critical question: is the performance bottleneck caused by the model's inability to generate high-quality videos that conform to physical laws (limited generative capacity), or by a deficiency in the model's underlying reasoning and logic? Current metrics fail to disentangle these two factors.

---

> ### Author Rebuttal · Authors · 2026-03-31
>
> Thank you for the thoughtful feedback and important questions regarding scaling behavior, data design, and evaluation.
>
> **Weakness 1a & Question 1: Validation on more architecture**
>
> We appreciate your suggestion to include additional architecture. Due to the limited rebuttal period, we include results on LTX-2.3 [1] below. LTX-2.3 is a multimodal dual-stream transformer that jointly generates audio and video via cross-attention between the two streams, and employs hierarchical upscaling for efficient high-resolution generation.
> |Model|In-Domain|Out-of-Domain|Overall|
> |:-|:-:|:-:|:-:|
> |0K (LTX2.3)|0.390|0.319|0.355|
> |100K|0.458|0.335|0.397|
> |200K|0.570|0.446|0.507|
> |300K|0.592|0.430|0.511|
> |400K|0.591|0.443|0.516|
> |500K (VBVR-LTX2.3)|0.580|0.453|0.516|
>
> These results further support that our key observation holds consistently across different architectures. Notably, performance improves rapidly at early scaling stages but begins to plateau beyond 200K–300K samples, with only marginal gains thereafter.
>
> **Weakness 1b & Question 2: Synthetic data cannot bridge the gap**
>
> Thank you for your insightful comment. We argue that the observed limitation cannot be explained solely by data diversity. Notably, the model fails to even fit the synthetic distribution: our in-domain test set consists of tasks structurally similar to the training data, yet performance still plateaus. If scaling were sufficient, the model should approach saturation on such in-domain data. The fact that it does not suggests a bottleneck beyond data quality, likely rooted in the current model architecture or learning paradigm.
>
> This observation aligns with prior work (e.g., Kang et al. [2]) in more constrained domains such as physics reasoning, and we extend it to a broader range of video reasoning tasks. We agree that incorporating richer real-world data is important. Therefore our claim is not that scaling is ineffective in general, but that, under current settings, scaling data alone is insufficient to close the gap in video reasoning, and we hope this motivates new exploration beyond scaling.
>
> In addition, we emphasize that knowledge learned from synthetic data remains useful. We evaluate VBVR-Wan2.2 on VBench, a real-world video benchmark, and observe clear improvements in controllability and consistency. In particular, Video-Text Camera Motion improves from $0.54 \to 0.65$, along with gains in Subject and Background Consistency. While Dynamic Degree decreases, we view this as an expected trade-off, as VBVR emphasizes strict adherence to logical prompts and precise local edits, which may limit unconstrained motion.
> |Model|Total Score|Video-Text Camera Motion|Video-Image Subject Consistency|Video-Image Background Consistency|Dynamic Degree|
> |:-|:-:|:-:|:-:|:-:|:-:|
> |Wan2.2-I2V|0.8816|0.5444|0.9752|0.9903|0.5285|
> |VBVR-Wan2.2|0.8835|0.6592|0.9804|0.9921|0.4106|
>
> **Weakness 2 & Question 3: Evaluation cannot decouple generative and reasoning capability**
>
> Thank you for this important question. We agree that fully disentangling generative capacity from underlying reasoning remains challenging. However, we clarify that our evaluation design provides a more controlled and informative signal compared to existing alternatives.
>
> First, VLM-based evaluators do not resolve this issue and may further entangle the two factors. Their judgments are sensitive to perceptual quality, prompt formatting, and other superficial cues, and are prone to hallucination, making it difficult to attribute errors to generation versus reasoning. In contrast, our rule-based evaluation provides a controlled setting where (i) reasoning constraints are explicitly defined, (ii) correctness is verified through deterministic rules, and (iii) confounding factors are minimized. In particular, VBVR leverages synthetic data to reduce distractions (e.g., texture, lighting, and aesthetics) that are orthogonal to reasoning. This enables more direct diagnosis of reasoning failures.
>
> Second, our task design goes beyond memorization. Many tasks require structured, multi-step constraint satisfaction and cannot be solved via simple pattern matching, making success indicative of genuine reasoning. Supporting this, recent work [3] (conducted on VBVR-Wan2.2) shows that the model exhibits implicit reasoning during diffusion: early denoising steps maintain multiple candidate solutions, which are progressively refined and pruned into a consistent outcome across diverse tasks, including real-world scenarios.
>
> Overall, while complete disentanglement remains an open problem, our framework provides a more controlled and interpretable evaluation of reasoning, offering valuable insights to the community.
>
> [1] HaCohen et al., LTX-2: Efficient Joint Audio-Visual Foundation Model, arXiv 2026.
>
> [2] Kang et al., How far is video generation from world model: A physical law perspective, arXiv 2024.
>
> [3] Wang et al., Demystifying Video Reasoning, arXiv 2026.

---

### Official Review · Reviewer_Xcz6 · 2026-03-13

**Soundness:** 3
**Presentation:** 3
**Significance:** 3
**Originality:** 2
**Overall Recommendation:** 5
**Confidence:** 3

**Summary:**

This paper introduces a remarkably large-scale video reasoning evaluation suite and dataset named VBVR. This work aims to address the data bottleneck restricting the complex reasoning capabilities of current large video models. The paper constructs a dataset comprising approximately 2 million samples, which is three orders of magnitude larger than existing benchmarks. The task design is grounded in human cognitive architecture, encompassing five dimensions: abstraction, knowledge, perception, spatiality, and transformation.

**Compliance With Llm Reviewing Policy:**

Affirmed.

**Final Justification:**

I believe the author has addressed all my concerns, and I've raised my score to 5. I hope the author will include the changes in the revised version.

**Key Questions For Authors:**

Seeing weaknesses

**Limitations:**

Seeing weaknesses

**Strengths And Weaknesses:**

Strengths:

- The scale of the dataset is massive. By employing parameterized generator code rather than relying on human annotation or direct generation by large models, the approach ensures low noise, high controllability, and exceptional scalability of the data.
- Eschewing the currently popular VLM-as-a-judge paradigm, the evaluation employs hard rules for automated scoring, ensuring that the assessment is completely reproducible and deterministic. Furthermore, its evaluation results exhibit a extremely high correlation with human preferences.
- Through qualitative and quantitative experiments, the paper clearly identifies the fundamental flaws of current video generation models when faced with long-horizon temporal consistency and strict logical constraints. The proposed insight that "controllability before reasoning" offers profound guiding value for the design of future video generation architectures.


Weaknesses:

- Currently, the VBVR data primarily consists of purely synthetic 2D animations based on simple geometric shapes, grids, and physics simulations. While this is highly conducive to decoupling and testing pure reasoning capabilities, the authors could additionally discuss: after fine-tuning on such highly simplified synthetic data, can the model's reasoning abilities effectively transfer to real-world, in-the-wild videos with complex textures and lighting?
- The experiment section notes that even as the data scale increases, the model's performance on In-Domain tasks eventually hits a bottleneck. The authors could explore this a bit deeper: what specific inductive biases are currently lacking in DiT or Transformer architectures to overcome this?
- I suggest briefly discussing your video data pipeline alongside a VLM data suite (such as Bee [1]). This would offer a broader perspective on how the field is solving the data quality bottleneck for open-source reasoning across images and videos.

[1] Zhang, Y., Ni, B., Chen, X. S., Zhang, H. R., Rao, Y., Peng, H., ... & Hu, S. M. (2025). Bee: A High-Quality Corpus and Full-Stack Suite to Unlock Advanced Fully Open MLLMs. arXiv preprint arXiv:2510.13795.

---

> ### Author Rebuttal · Authors · 2026-03-31
>
> We want to thank you for the constructive feedback and for recognizing the "massive scale," "reproducible evaluation," and the "profound guiding value" of our controllability before reasoning insight. Below we address the specific concerns raised.
>
> **Weakness 1: Transfer from 2D synthetic data to real-world videos**
>
> Thank you for your insightful comments. We appreciate your recognition that synthetic data is employed to decouple reasoning from visual distractions. Despite being predominantly synthetic, we observe that the VBVR-Dataset demonstrates strong transferability to real-world scenarios.
>
> First, we evaluate the VBVR-Wan2.2 model on VBench, a benchmark designed for real-world video generation. As shown in the table below, our model exhibits clear improvements in controllability and consistency. Specifically, we observe gains in Video-Text Camera Motion ($0.54 \to 0.65$) as well as in Subject and Background Consistency. Meanwhile, there is a decrease in Dynamic Degree, which we consider an expected trade-off. This arises because the VBVR-Dataset emphasizes strict adherence to logical prompts and precise local edits, which can limit unconstrained motion.
>
>
> |Model|Total Score|Video-Text Camera Motion|Video-Image Subject Consistency|Video-Image Background Consistency|Dynamic Degree|
> |:-|:-:|:-:|:-:|:-:|:-:|
> |Wan2.2-I2V|0.8816|0.5444|0.9752|0.9903|0.5285|
> |VBVR-Wan2.2|0.8835|0.6592|0.9804|0.9921|0.4106|
>
>
> Second, we provide additional qualitative analysis at https://github.com/engineerAnonymous/2026_VBVR_rebuttal, where the model demonstrates consistent constraint-aware planning. For example, it avoids invalid trajectories such as “wall penetration” in maze-like environments and shows stronger prompt following capabilities. This behavior suggests that the learned reasoning extends beyond superficial visual patterns and generalizes to structurally valid decision-making.
>
> **Weakness 2: Architectural Bottlenecks and Inductive Biases**
>
> Recent work such as Demystifying Video Reasoning [1], conducted on VBVR-Wan2.2, suggests that video diffusion models do not merely memorize patterns, but instead exhibit implicit reasoning during the denoising process. Specifically, early diffusion steps maintain multiple candidate solutions, which are progressively refined and pruned in later steps toward a consistent outcome.
> However, the current tasks remain highly challenging, and existing video generation models typically rely on text encoders (e.g., T5 [2]) that may struggle to capture the nuanced details in task prompts that are critical for accurate understanding. This limitation can hinder downstream reasoning performance. With the emergence of unified understanding and generation models (e.g., OmniGen2 [3], Bagel [4]), we believe it is timely to equip video reasoning models with stronger understanding components, such as vision-language models (VLMs), to better capture complex semantics and support more robust reasoning.
>
> **Weakness 3: Compare with VLM data pipeline**
>
> Thank you for this insightful suggestion. In the revised version, we will expand the Related Work section to more clearly position VBVR alongside recent VLM data suites such as Bee [5]. While Bee represents a significant step in scaling multimodal SFT via MLLM-driven data enrichment (e.g., using open MLLMs for short-form CoT and proprietary models for long-form CoT on image–text pairs), VBVR adopts a different approach tailored for video reasoning.
>
> Rather than relying on model-based data curation, our approach emphasizes procedural generation. Specifically, VBVR derives deterministic ground truth from carefully designed, manually implemented simulators, producing physically grounded and fully controllable supervision over rendered videos. This ensures high-quality reasoning content, where temporal dynamics and physical constraints are explicitly defined and guaranteed to be correct.
>
> Furthermore, VBVR-Bench employs rule-based, task-specific evaluators instead of a VLM-as-a-judge paradigm. Evaluation outcomes are computed as deterministic functions of the generated videos and task constraints, ensuring reproducibility while avoiding reliance on potentially inconsistent generative judges.
>
> [1] Wang et al., Demystifying Video Reasoning, arXiv 2026.
>
> [2] Raffell et al., Exploring the limits of transfer learning with a unified text-to-text transformer, Journal of machine learning research, 21(140), 1-67, 2020.
>
> [3] Wu et al., Omnigen2: Exploration to advanced multimodal generation, arXiv 2025.
>
> [4] Deng et al., Emerging properties in unified multimodal pretraining, arXiv 2025.
>
> [5] Zhang et al., Bee: A High-Quality Corpus and Full-Stack Suite to Unlock Advanced Fully Open MLLMs, arXiv 2025.

---

> > ### Author Rebuttal · Reviewer_Xcz6 · 2026-04-04
> >
> > I believe the author addressed my concerns, and I raise my rating. Hopefully, the authors can incorporate the above improvements into the reversion.

---

> > > ### Author Response · Authors · 2026-04-04
> > >
> > > We sincerely thank the reviewer for the thoughtful and constructive feedback. We also appreciate the reviewer’s positive reassessment and increased rating. The additional suggestions provided are valuable, and we will incorporate them into the final version of the paper.

---

### Decision · Program_Chairs · 2026-04-30

**Decision:**

Accept (regular)

**Comment:**

All reviewers agree that the scale and diversity of the dataset constitute a key strength of the work.

The reliance on synthetic data was initially raised as a concern. However, the rebuttal provides additional results on real-world data, which alleviates this issue and reduces it to a minor concern.

The use of rule-based scorers for evaluation is viewed inconsistently across reviewers: some (e.g., Reviewers Xcz6 and JC8N) consider it a strength, while others (e.g., Reviewer v9vY) see it as a limitation.

Overall, based on the balance of reviews, the arguments in favor of acceptance outweigh those for rejection. AC recommends for weak accept.